# Ice core evidence for major volcanic eruptions at the onset of Dansgaard-Oeschger warming events

Johannes Lohmann[1] and Anders Svensson[1]

[1]Physics of Ice, Climate and Earth, Niels Bohr Institute, University of Copenhagen, Denmark

**Correspondence:** Johannes Lohmann (johannes.lohmann@nbi.ku.dk)

**Abstract.** While a significant influence of volcanic activity on Holocene climate is well-established, an equally prominent role of major eruptions on the climate variability and regime shifts during the Quaternary glacial cycles has been suggested. Previous statistical assessments of this were challenged by inaccurate synchronization of large volcanic eruptions to changes in past climate. Here, this is alleviated by combining a new record of bipolar volcanism from Greenland and Antarctic ice cores with records of abrupt climate change derived from the same ice cores. We show that bipolar volcanic eruptions occurred significantly more frequently than expected by chance just before the onset of Dansgaard-Oeschger events, the most prominent large-scale abrupt climate changes of the last glacial period. Out of 20 abrupt warming events in the 12-60 ka period, 5 (7) occur within 20 (50) years after a bipolar eruption. We hypothesize that this may be a result of the direct influence of volcanic cooling on the Atlantic meridional overturning circulation, which is widely regarded as the main climate subsystem involved in Dansgaard-Oeschger cycles. Transitions from a weak to a strong circulation mode may be triggered by cooling in the North Atlantic, given the circulation is close to a stability threshold. We illustrate this suggestion by simulations with an ocean-only general circulation model forced by short-term volcanic cooling. The analysis presented suggests that large eruptions may act as short-term triggers for large-scale abrupt climate change, and may explain part of the variability of Dansgaard-Oeschger cycles. While we argue that the bipolar catalogue used here covers a sufficiently large portion of the eruptions with the strongest global climate impact, volcanic events restricted to either the northern or southern hemisphere may likewise contribute to abrupt climate change.

## 1 Introduction

Volcanic eruptions have been shown to be a major driver of climate variability (Robock, 2000; Schurer et al., 2013, 2014; Sigl et al., 2015; Swingedouw et al., 2017; Mann et al., 2021). Besides the warming trend due to greenhouse gas emissions, their impact on global mean temperature in historical climate simulations is the only feature that exceeds the ensemble uncertainty in the latest generation of climate models (Tokarska et al., 2020). However, the potential impact of individual very large volcanic eruptions on global climate is not very well constrained due to the very small number of such events to occur since instrumental climate observations began. It is not known whether large volcanic eruptions can drive the evolution of climate on longer time scales, and whether the climate's response can go beyond a linear short-term relaxation after sudden cooling related to stratospheric sulfate aerosols.

Climate variability on the millennial time scale is mostly associated with the Dansgaard-Oeschger (DO) and Heinrich events of the last glacial period. The former consisted of about 30 cycles (Dansgaard et al., 1993) where the cold glacial climate in the Northern Hemisphere (Greenland stadial periods, GS) was interrupted by abrupt warmings of 8-15 K within a few decades (Kindler et al., 2014), followed by prolonged periods of milder but gradually decreasing temperatures (Greenland interstadials, GI). This was often followed by another abrupt transition back to stadial conditions. By synthesizing proxy data from different archives, much progress has been made in unraveling the mechanisms behind these large-scale climate oscillations (Dokken et al., 2013; Lynch-Stieglitz, 2017; Pedro et al., 2018; Sadatzki et al., 2019). At the same time, long simulations with realistic Earth system models became possible, some of which show unforced oscillations of the climate that are very similar to DO cycles (Zhang et al., 2021; Klockmann et al., 2020, 2018; Brown and Galbraith, 2016; Vettoretti and Peltier, 2016). Still, a consensus regarding the concrete drivers, if any, that lead to transitions in between stadials and interstadials has not been achieved yet.

One challenge for existing hypotheses concerns the irregular occurrence times of DO events. While averaging roughly 1500 years, the individual stadials and interstadials that comprise the DO cycles can last anywhere from less than a century up to ten millennia (Rasmussen et al., 2014). The most realistic model simulations show rather regular self-sustained oscillations of the climate (Vettoretti and Peltier, 2016). However, they currently do not include important factors, such as interactive ice sheets, carbon cycle, and external insolation and volcanic forcing, which might change the nature of the oscillations. This makes it so far difficult to judge whether the simulations are fully consistent with the observed properties of DO cycles: On the one hand, the statistics of the time elapsed before a DO warming transition, as well as the statistics of the durations of the transitions themselves, are consistent with a purely stochastic driver (Ditlevsen et al., 2005; Lohmann and Ditlevsen, 2019). On the other hand, there is evidence for external influences of insolation, atmospheric $CO_2$ and global ice volume on the lengths of the cycles (Schulz, 2002; Buizert and Schmittner, 2015; Mitsui and Crucifix, 2017; Kawamura et al., 2017; Lohmann and Ditlevsen, 2018, 2019), as well as for deterministic features in the data that allow for a prediction of the occurrences of DO events with significant skill (Lohmann, 2019). Nevertheless, such predictions do not perfectly explain the full variability of the occurrence times and leave room for stochastic drivers that influence the timing of event occurrence. One such driver could be large volcanic eruptions.

A causal relationship between volcanic eruptions and abrupt climate change has been suggested before. The initiation of prominent climate transitions, such as the termination of the last glacial period, the onset of the Younger Dryas cold event, as well as the transition from the Medieval Climate Anomaly to the Little Ice Age, have been proposed to be caused by large volcanic eruptions (Schleussner et al., 2015; McConnell et al., 2017; Baldini et al., 2018; Abbott et al., 2021). While it is difficult to substantiate causality for individual events beyond doubt, a statistical analysis of re-occurring abrupt climate change events has the potential of establishing a systematic link to external drivers such as volcanic eruptions. This requires records of large volcanic eruptions and climate change events that are as complete as possible, along with a precise age control to tie the eruptions to climatic changes. Those requirements have been challenging for previous studies.

By comparing the largest well-known and absolutely dated volcanic eruptions to climate changes identified in ice cores it was found that within dating uncertainties large Northern Hemisphere (NH) eruptions tend to cluster around the abrupt

cooling phases of DO cycles, whereas large Southern Hemisphere (SH) eruptions might be associated with DO warmings (Baldini et al., 2015). However, the absolute age uncertainties of the layer-counted ice core chronologies as well as those of the radiometrically dated eruptions are typically of the same order of magnitude and grow to more than a millennium during the last glacial period. Similarly, eruptions recorded in an Antarctic ice core and the abrupt cooling events of DO cycles in a Greenland ice core were reported to be clustered closer to another than could be expected by chance (Bay et al., 2004). When allowing for a volcanic lead time of 2 kyr, Bay et al. (2004) also reported a clustering of volcanic eruptions and the abrupt DO warming transitions. Still, in this case there were multi-century synchronization uncertainties, limiting the confidence in a direct effect of the eruptions on the climate and making it impossible to judge the temporal order of eruptions and climate change. Additionally, by defining volcanic eruptions from a single ice core this study inevitably included smaller, local eruptions with limited climatic impact.

The present study overcomes both of these issues by using a recently published record of bipolar volcanic eruptions identified in polar ice cores in the interval 11-60 ka, and the associated bipolar volcanic match points in the individual ice cores (Svensson et al. (2020), SVE20 hereafter). First, by concentrating on volcanic eruptions that led to significant sulfate deposition at both poles, as seen in Antarctic and Greenland ice cores, all eruptions can be expected to be above a certain threshold in magnitude, and are thus likely to have had large climatic impacts. We cross-check this assertion by taking into account a recently published continuous sulfate deposition record of volcanic eruptions during the last glacial period (Lin et al. (2022), LIN22 hereafter), which allows an estimation of the eruption magnitudes. Compared to the previous study by Bay et al. (2006) with a very sparse record that made it difficult to obtain statistical conclusions, the data set employed here contains a much larger number of eruptions. The bipolar matches have furthermore been obtained in a much more reliable way due to parallel layer counting in Greenland and Antarctica in between events. Second, the timing uncertainties between eruptions and climate transitions are greatly reduced by combining this reliable matching of volcanogenic sulphur depositions in Antarctic and Greenland ice cores with a determination of climate transitions from high-resolution isotopic records of the same well-synchronized Greenland ice cores. This allows us to assess the potential occurrence of volcanic eruptions leading up to abrupt climate change with decadal precision. Specifically, we test the temporal proximity of $N = 82$ bipolar volcanic eruptions to $M = 20$ DO events against a null hypothesis of random and uncorrelated occurrences of volcanic eruptions.

## 2 Methods and Materials

### 2.1 Greenland high-resolution isotope records and stack

We consider high-resolution $\delta^{18}O$ records of four deep Greenland ice cores, as well as a stack derived from these. The records have been measured at different depth resolutions, where each measurement was performed on bulk material of contiguous depth intervals. The measurements are thus not point samples, but averages over contiguous intervals. We use the $\delta^{18}O$ records from the NGRIP ice core (NGRIP Members, 2004; Gkinis et al., 2014), as well as the GRIP (Johnsen et al., 1997), GISP2 (Stuiver and Grootes, 2000) and NEEM (Rasmussen et al., 2013; Gkinis et al., 2021) ice cores. All cores are synchronized to the annual layer-counted Greenland Ice Core Chronology 2005 (GICC05) (Svensson et al., 2006, 2008; Rasmussen et al.,

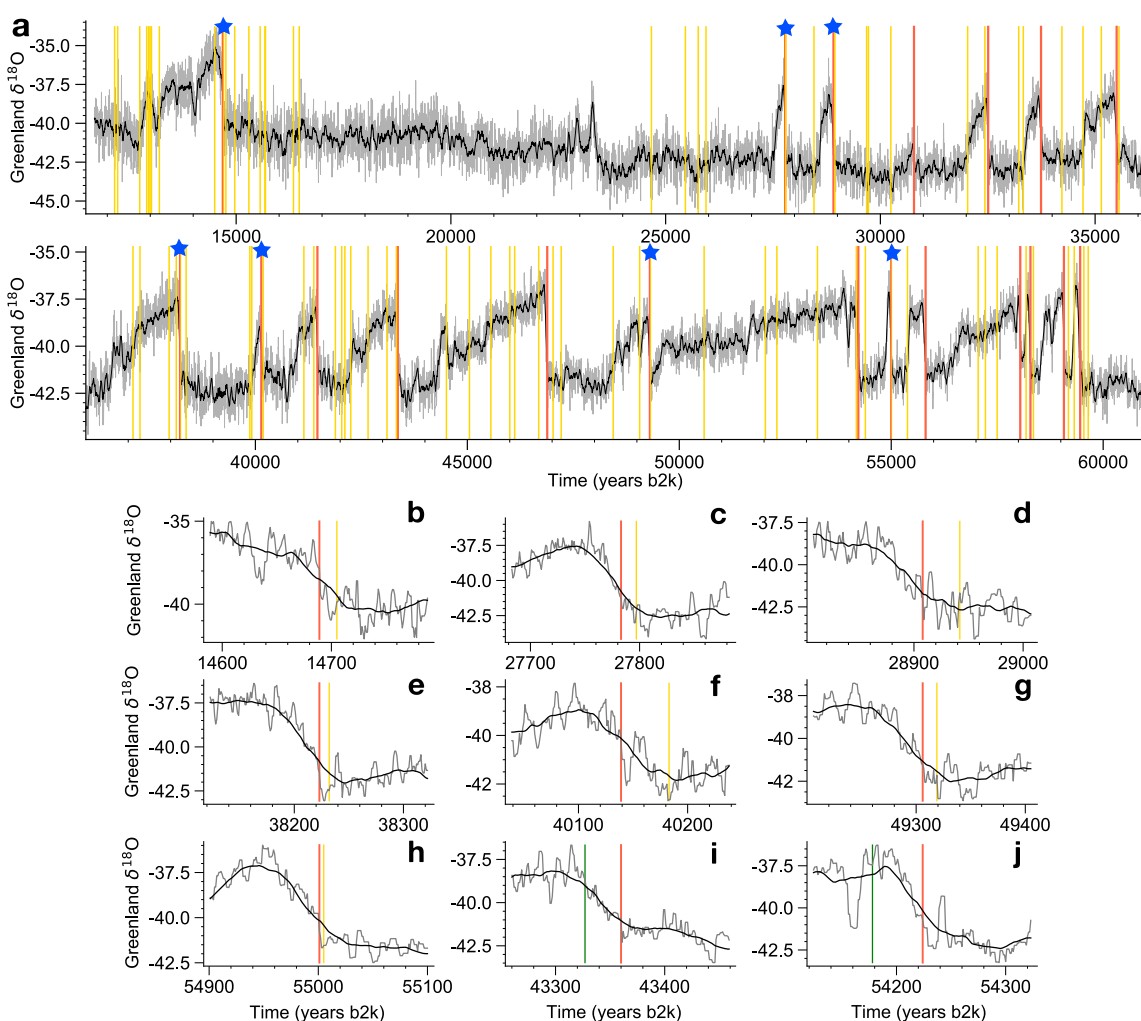

**Figure 1.** Records of abrupt climate change and bipolar volcanism during the last glacial period. **a** High-resolution Greenland $\delta^{18}O$ stack (gray, see Sec. 2.1) and 50-year low-pass filtered stack (black), together with the estimated onsets of the DO warmings (red, this work) and the bipolar volcanic eruptions (yellow, SVE20). For the interval 16.5-24.5 ka no bipolar volcanic eruptions have been identified as the ice cores are difficult to synchronize. The blue stars indicate instances where a volcanic eruption occurs within 50 years prior to the DO warming. The corresponding segments are shown magnified in panels **b-h**. For comparison, panels **i-j** show the only two instances where a bipolar eruption (marked in green) occurs within 50 years after the DO warming onset.

2013; Seierstad et al., 2014). The $\delta^{18}O$ records are processed in the following way. The midpoints of the depth intervals are interpolated linearly to the GICC05 time-depth scale, yielding an unequally spaced time series. Then, this series is oversampled to a 1-year equidistant grid using nearest-neighbor interpolation. In this way, the nature of the measurements as contiguous depth averages and the original measurement values are preserved, and all records are placed on the same equidistant time grid. Finally, we average the four ice cores to obtain a stack with significantly reduced high-frequency noise. We estimate the DO

onsets both from the stack and the individual ice cores except for GISP2, due to the comparably low resolution of the raw data in the stadials: Starting at GS-2, the sample resolution decreases from 8 years to around 30 years at GS-15.2. For comparison, the stadial resolutions for the same time periods of NGRIP, GRIP and NEEM are 3 to 6, 4 to 9 and 4 to 8 years, respectively. The time period we consider spans from 11,700 years b2k (years before 2000 AD) to 60,000 years b2k (GS-18). We do not consider the DO events 2.1 and 2.2, as no bipolar matching has been obtained for this interval.

## 2.2 Ice core records of last glacial volcanism

The primary volcanic data set used in this study is the last glacial record of bipolar volcanism obtained by SVE20. It contains $N = 82$ volcanic eruptions where a significant sulfate deposition could by identified synchronously in Greenland and Antarctic ice cores. With very few exceptions, these eruptions are all of unknown source. While the data set spans the interval 11.7 - 60 ka, there is a data gap from 16.5 - 24.5 ka due to the difficulties in synchronizing the Greenland and Antarctic cores as a result of the high impurity noise levels during the late glacial. Discarding this gap, this yields a duration of $\Delta T = 40,300$ years investigated in this study.

In order to analyze the statistics of this data set, and to constrain how many bipolar eruptions of similar magnitude may be missing from it, we also consider the record by LIN22 covering the same period obtained by an automated method to detect volcanic eruptions from continuous sulfate measurements in ice cores from Greenland and Antarctica. From these measurements, Lin and co-workers derived two separate lists of volcanic eruptions and their total sulfate deposition, considering eruptions detected individually in either Greenland or Antarctica, respectively. These lists contain eruptions with sulfur deposition exceeding 10 kg/km$^2$ and 20 kg/km$^2$ in Antarctica and Greenland, respectively, which corresponds to roughly half of the deposition of the 1815 AD Tambora eruption. Importantly, the lists also contain the bipolar eruptions from SVE20. For these bipolar eruptions, from the sulfur deposition at both poles Lin and co-workers computed the stratospheric aerosol loading as a measure for the global climate forcing, which we use to gauge how large the bipolar eruptions are compared to more well-known recent eruptions (Sec. 2.4). Further, based on the relative deposition in Greenland versus Antarctica, Lin and co-workers give rough estimates for the eruption latitude, which is discussed in more detail in Sec. 3.7.

The eruptions remaining from the data sets of LIN22 after removing the bipolar eruptions from SVE20 will be called the *unipolar* data sets hereafter. In the interval 11.7 - 60 ka minus the bipolar data gap 16.5 - 24.5 ka, this yields 740 and 498 eruptions in the Greenlandic and Antarctic data set, respectively. By comparing the sulfate depositions of these unipolar eruptions to the bipolar ones, we can derive an upper limit estimate on how many unidentified bipolar eruptions of similar magnitude may be missing from the SVE20 data set, assuming the LIN22 data set would not contain any regional (relatively close to the ice core sites) eruptions with a large (uni-)polar deposition but a small global impact.

## 2.3 DO event onset determination

We determine the timing of the DO warming onsets from high-resolution $\delta^{18}$O records of individual ice cores, as well as a stacked record. For each DO cycle, we detect when the data start to deviate significantly upwards from the stadial mean before the abrupt transitions. The method is explained in detail in Appendix A. The identified onsets for the isotopic stack are shown

together with the isotopic record and the closest bipolar volcanic eruption preceding the onset in Fig. S1. The estimated onset times in the individual cores are compared to the stack in Fig. S2, where the relative onset timings are given with respect to the earliest of the four independent onset determinations.

This comparison allows us to assess the reliability of the onset estimates from the stack and obtain an estimate for the temporal uncertainty. The onsets are very consistent for the events 3, 4, 5.2, 6, 7, 8, 10, 16.1, 16.2, where the range of the four independent onset estimates is 25 years or less. For the events 1, 5.1, 9, 11, 13, 14, 15.1, 17.1 and 17.2, the stack onset clusters with 2 other ice cores within less than 20 years, while the onset of the remaining ice core is further apart from this cluster and can be considered an outlier. Some outliers may be due to imperfect synchronization, as in the case of events 5.1 and 17.1. Here, the match points used to synchronize NEEM to the GICC05 time scale are spaced several hundred years before and after the DO onset, and thus there is likely an offset in the timing at the onset. In other cases, the shape of the onset in one of the ice cores is not as abrupt as in the others, or is very noisy, which leads to a late detection with our algorithm. There can also be an early detection when there is a large spike in the high-frequency noise before the onset, as with GI-14 in NEEM. For the remaining events 12 and 15.2, the DO transition appears to consist of two steps in all cores, and due to the high noise level our algorithm is not able to detect the first step in NGRIP, which we argue to be the true onset. To summarize, the onsets derived from the stack should be considered most representative. First, this is because the stack is better suited to define the onsets due to its improved signal-to-noise ratio. Second, the stack onsets are consistent with the timing estimates from the individual cores. Discarding outliers, we find that the spread of the individual onset estimates (including the estimates from the stack), i.e., the difference of the earliest and latest onset estimate, is 12.9 years on average. This can serve as a good estimate of the uncertainty of the onsets determined from the isotopic stack. It is therefore possible to assess the climatic repercussions of volcanic eruptions to a decadal-scale.

## 2.4 Occurrence statistics of bipolar eruptions

To test whether DO events preferentially occur shortly after volcanic eruptions, we need to know the occurrence rate of the latter over time and check the validity of our null hypothesis. The SVE20 data set covers the time interval from 11.7 - 60 ka minus the data gap from 16.5 - 24.5 ka, which gives a total of 20641 years of stadial condition with 42 volcanic eruptions and 19659 years of interstadial condition with 40 volcanic eruptions. This yields remarkably similar occurrence rates of bipolar eruptions of $\lambda = 2.0348$ eruptions per kyr for stadials and $\lambda = 2.0347$ for interstadials, corresponding to a return period of close to 500 years. In stadial periods Greenland ice cores show a much higher level of the different impurity signals, which is thought to be a result of changes in atmospheric circulation along the transport route, changes in wind at the source, and changes in the hydrological cycle, as compared to the interstadial periods (Ruth et al., 2003). While the higher background level in the stadial sulfate records certainly includes many smaller regional eruptions, as well as small bipolar eruptions, it also has the potential to impede the detection of the large bipolar eruptions that are considered in this study. However, despite the higher stadial background sulfate levels, the similar occurrence rates indicate that there is no systematic undercounting of eruptions during the stadials compared to interstadials.

Since the SVE20 data set has not been obtained using an automated method, it remains a concern that there may be a significant number of bipolar eruptions of similar magnitude missing, and that there may be a systematic bias towards a preferred detection of eruptions close to abrupt DO transitions. The bipolar matching in SVE20 was achieved by simultaneous layer-counting in Greenland and Antarctic cores. This layer-counting was not fully continuous, but did still cover almost the entire period, with 16.5% of the record missing (see Table S2 in SVE20). Further, the method of bipolar matching is by construction not continuous, but entails the identification of patterns of typically 3-5 eruptions in both Greenland and Antarctica, which can be matched by the number of annual layers found in between eruptions. However, given an average spacing of the eruptions of 500 years, one can easily see that these patterns would cover the majority of the entire time period even if only patterns centered around the 20 DO onsets were chosen. Still, even if the entire period was investigated as thoroughly as possible by SVE20, some large bipolar eruptions may have been missed within the covered time intervals, and we need to show that a potential undercounting of bipolar eruptions does not compromise the suitability of the data set and the null hypothesis, as well as the robustness of our results. This is addressed in the following by comparing the magnitude and return period observed in the SVE20 data set to well-known historic eruptions, as well as by deriving an upper estimate for the number of potentially unidentified bipolar eruptions of the same magnitude and testing the robustness of our results against it (Sec.'s 3.2 and 3.5). The concern of a systematic bias of the bipolar catalogue of SVE20 towards periods of abrupt transitions is addressed in Sec. 3.1, as well as in Sec. 3.2, where the magnitudes of eruptions close to DO onsets is compared to eruptions elsewhere.

In comparison to the more than 80 bipolar eruptions obtained for the last 2500 years by Sigl et al. (2015), as well as the more recent analysis of ice core records covering the entire Holocene yielding one eruption with bipolar imprint every 35 years (Sigl et al., 2022), the eruptions considered here are indeed quite sparse. However, this is expected because of the layer thinning of ice cores with depth, the lower accumulation rates during the glacial period, and due to the much higher background levels of the impurity signals during the last glacial (Mayewski et al., 1997; Schüpbach et al., 2018). Therefore, the sulfate records of the Holocene allow for the detection of much smaller bipolar eruptions. We thus only expect eruptions of very large magnitude to be present in our data set. In the data set of Sigl et al. (2015), five eruptions during the last 2500 years were found to be larger than the 1815 AD Tambora eruption in terms of their bipolar sulfate deposition. The magnitude of these 1-in-500 year events may be compared to the bipolar sulfate deposition of the glacial eruptions considered here, which has been estimated by LIN22 (see also Sec. 2.2). It is found that 69 of the bipolar eruptions have a stratospheric aerosol loading larger than the Tambora eruptions. Thus, most eruptions in the SVE20 data set fall into the category of 1-in-500 year events in terms of their magnitude. This corresponds very well to the fact that the occurrence rate of the eruptions is indeed once in 500 years, as shown above. Another well-known candidate to validate the eruption frequency in the SVE20 data set is the 1257 AD Samalas eruption. There are 29 eruptions in the SVE20 data set that, according to the estimate from LIN22, have a larger aerosol loading than the Samalas eruption. This subset of eruptions yields a return period of 1,390 years. From the data in Sigl et al. (2015), the Samalas eruption emerges as either the largest eruption of the last 2,000 years or the second largest eruption of the last 2,500 years. Thus, our return period is also consistent with recent ice core estimates of the eruption frequency of these even larger eruptions.

An independent comparison with records of volcanism derived from sources other than ice cores is desirable. For example, the compiled record of Rougier et al. (2018) yields return periods for M6 and M7 eruptions of 110 and 1200 years, respectively, where M7 signifies a binned range of eruption magnitudes containing both Tambora and Samalas. However, these numbers are based on estimates of the erupted mass of known eruptions, which is not directly calibrated to the polar sulfur deposition in the ice cores. Further, even though the estimates of M6 and M7 by Rougier et al. (2018) are only based on data of the last 2,000 years, the data is still likely incomplete compared to the ice core record because it only contains eruptions of known source. In the study by Sigl et al. (2015), 12 out of the 25 largest bipolar eruptions of the last 2,500 years could not be matched to an eruption of known source. Thus, a quantitative comparison of the data by SVE20 and LIN22 is currently only feasible with respect to the younger ice core record. This comparison, as reported above, indicates that the SVE20 data set is relatively complete, given that the frequency of eruptions of this magnitude and the associated polar sulfur deposition has remained approximately constant. By saying the data set is relatively complete, we mean that it covers a sufficiently large portion of the bipolar eruptions above a certain threshold in magnitude, which corresponds to eruptions with return periods of 1 in 500 years and larger. In contrast, we do not mean that it represents a complete catalogue of all bipolar eruptions of any size that could be detectable in more highly resolved and better synchronized ice core records, such as during the Holocene (Sigl et al., 2022). An estimate of an upper bound on the number of large bipolar eruptions potentially not identified by SVE20 is given in Sec. 3.2.

In the following, we test whether the timings of the bipolar eruptions are consistent with a stationary Poisson process. First, we need to check whether the distribution of waiting times $t$ in between eruptions is consistent with an exponential distribution with cumulative probability $P_\lambda(T \geq t) = 1 - e^{-\lambda t}$. This is indeed the case, as seen by an Anderson-Darling (Kolmogorov-Smirnov) test with $p = 0.60$ ($p = 0.66$). The corresponding empirical distribution function along with $P_\lambda(T \geq t)$ using $\lambda = 2.0348$ is shown in Fig. S3a. Second, we confirm that the memoryless property of the waiting times $t$ holds. This is achieved by a two-tailed bootstrap hypothesis test on the Spearman correlation of consecutive waiting times, yielding $p = 0.871$ for the data correlation of $r_S = 0.019$. Finally, we test the assumption of a constant rate $\lambda$ over time by dividing the volcanic record into short contiguous segments of $\Delta T$ years and testing whether the number $n$ of eruptions in each of them is consistent with a Poisson process at fixed $\lambda = 2.0348$. This is done by calculating the cumulative Poisson distribution function

$$P(N(\Delta T) \leq n) = \sum_{i=0}^{n} \frac{(\lambda \Delta T)^n}{n!} e^{-\lambda \Delta T}. \tag{1}$$

Choosing $\Delta T = 2$ kyr, except for slightly shorter values at the margins of the investigated time intervals, we find that 2 out of 21 segments lie outside of the 90% confidence region marked by $P = 0.05$ and $P = 0.95$ (for full results see Fig. S3b). These 2 segments occur in the youngest part of the record and could be related to a previously reported increased volcanic activity during the deglaciation (Zielinski et al., 1997). The recent analysis of ice core data by LIN22 (see Fig. 4 therein) suggests that the increased activity during the deglaciation is confined to eruptions with a large sulfur deposition in Greenland. But it has not been established whether this is statistically significant, and we cannot reject that it simply reflects the higher chances of detecting volcanic signals due to better signal preservation in the younger parts of the record.

For the case of the bipolar eruptions considered here, we expect 2.1 false positives when testing 21 independent hypotheses at 90% confidence. Thus, even with the 2 segments in question, the data do not provide evidence to conclude that the eruption

rate changed significantly over time. Instead, it is consistent with a stationary Poisson process. We performed the same analysis by choosing the individual stadials and interstadials as data segments, instead of segments at fixed $\Delta T$. The results are shown in Fig. S4, and again we see no evidence to suggest that the eruption rate is not constant over time, or biased for stadials or interstadials. Thus, while there may be a higher frequency of eruptions towards the deglaciation, in the context of the whole time interval 12-60ka this may have occurred by chance and it is not warranted to take the higher frequency into account in our null hypothesis. Still, we test the robustness of our results against a higher eruption frequency (Sec. 3.5), and since it only is a short segment of the whole time period, the potentially enhanced volcanic frequency during the deglaciation is unlikely to change our conclusions.

Finally, we remark that the missing data gap does not influence the results reported here. Given that the frequency of eruptions is approximately constant, there is no need for a continuous time period since only the rate of eruptions enters the calculations, and the latter does not change significantly when choosing different points where the record is cut. More concretely, the amount of 'empty' space without DO events that arises at the boundaries of the data segments (or gaps) does not make a difference in our analysis. This is because we are asking how often a DO event is preceded by an eruption, and not how likely it is that an eruption triggers a DO event. Only in the latter case, the amount of 'DO-empty' space at the beginning and end of a data interval would influence the results. Thus, assuming the statistics of eruptions and the statistical relationship in between DO events and eruptions were not significantly different in the data gap, our results are not biased by the latter.

## 2.5 Global ocean model simulations

Using a global ocean model, we investigate whether the statistical connection of large volcanic eruptions and DO cycles revealed here can be explained by a triggering of transitions of the AMOC due to short-lived volcanic cooling. These model simulations will serve as a proof of principle and not a comprehensive study of the influence of volcanic forcing on the DO atmosphere-ice-ocean variability under realistic glacial boundary conditions. The latter is very challenging, since ultimately the main drivers of DO events remain only partially understood, and only few models are able to reproduce climate variability similar to DO cycles. Thus, our simulations merely illustrate a hypothesis to be tested by further realistic modeling studies.

Here we use the ocean model *Veros* (Häfner et al., 2018), which is a primitive equation finite-difference model set up in a global present-day configuration with prescribed present-day atmospheric forcing. These present-day conditions differ significantly from the boundary conditions of the last glacial period, and consequently the strength and stability of the AMOC may be different. However, the true strength and stability of the glacial AMOC is not well-constrained and was varying significantly over time. Instead of trying to derive consistent glacial boundary conditions for which a desired dynamical AMOC regime emerges, we tune the boundary conditions such that the system shows the kind of AMOC instability that is widely believed to be responsible for the transitions in between stadial and interstadial conditions during DO cycles. Specifically, we use a suitable change to the salinity boundary condition via increased North Atlantic (NA) freshwater input under which the model supports multiple stable states of the AMOC, i.e., states of vigorous as well as collapsed overturning. These states serve as analogues to the stadial and interstadial states of the last glacial. Tipping in between these states of the model has been investigated previously by Lohmann and Ditlevsen (2021). Compared to the latter study, we made minor changes to the model

configuration to improve realism and numerical properties. First, while we use the same resolution with 90 longitudinal and 40 latitudinal grid cells (where the latitudinal resolution increases from 5.3$^o$ at the poles to 2.1$^o$ at the equator), as well as 40 vertical layers (increasing in thickness from 23m at the surface to 274m at the bottom), the interpolation of the bathymetry is done using a Gaussian filter with different bandwidth, leading to more realistic land masses. Second, present-day ERA-40 wind stress forcing is employed. The heat and freshwater exchange with the atmosphere is modeled by boundary conditions, where the sea surface temperature and salinity are relaxed towards present-day climatological fields (Uppala et al., 2005) within a given relaxation time scale. Choosing a longer time scale for the salinity compared to the temperature relaxation enables the positive salt advection feedback, which is thought to be responsible for the existence of multiple stable AMOC regimes (Weijer et al., 2019). Here we use a relaxation time scale of 2 years for salinity and 30 days for temperature. Note that the model does not include an active sea ice or atmospheric component. The presence of these components may significant alter the stability of the AMOC and its response to volcanic eruptions. Our proposed mechanism is thus contingent on the sea ice and atmospheric response to not overrule the AMOC bi-stability and the dense-water formation in the North Atlantic as a volcanic trigger of an AMOC transition. A further consequence of the ocean-only framework is that no fast climatic processes are included. Thus, transitions in between different circulation states are not as abrupt as they may be expected due to the presence of fast, amplifying feedbacks such as abrupt changes in sea ice extent (Li et al., 2005). For further details of the ocean model, we refer the reader to Lohmann and Ditlevsen (2021), and Häfner et al. (2018).

We start with a control run using present-day initial and boundary conditions. Given the coarse resolution, this yields a reasonably realistic large-scale ocean circulation, with a maximum AMOC strength of about 18 Sv, and a maximum strength of the North Atlantic and subpolar gyres of 45 Sv and 25 Sv, respectively (see Fig. S9). Next, we gradually introduce a freshwater forcing in the NA, in order to find the tipping point corresponding to an AMOC collapse and to establish a regime of bistability, where AMOC transitions in between alternative stable states could be triggered by volcanic forcing. Because of the salinity relaxation boundary conditions, an added constant freshwater flux (negative salinity flux) is equivalent to a change of the salinity forcing field $\phi_i$ at grid cell $i$. We thus gradually change the forcing field in the grid cells between 296$^o$W to 0$^o$W and 50$^o$N to 75$^o$N (see Fig. S8). This corresponds to an area of roughly $A = 1.5$ mio. km$^2$. Assuming a constant reference salinity of $S_{ref} = 35$ gkg$^{-1}$, a change in the salinity forcing field can be converted to an equivalent total freshwater forcing $F = \phi A S_{ref}^{-1}$ in Sv (1 Sv $\equiv 10^6$m$^3$s$^{-1}$). We ramp up the freshwater anomaly $F$ in small increments, where one increment consists of a 200-y linear increase of $F$, followed by a 100-y relaxation period at constant $F$. At a value of $F = 0.49$ Sv, the anomaly was ramped back down to 0 using the same increments. The total simulation time of this transient experiment was 49,200 years. The AMOC collapse occurred at a higher value of $F$ compared to the resurgence. This reveals hysteresis and bistability, which serves as the basis to test our hypothesis of a direct trigger of transitions in between coexisting states of the AMOC as a response to volcanic cooling. To do this, starting from long equilibrium simulations at fixed values of $F$ within the bistability region, we perform simulations where a short-lived volcanic cooling is introduced by temporarily changing the values of the atmospheric temperature boundary condition. For the simulations reported here, all eruptions were initiated in the autumn, but the results are not sensitive to the season of eruption (see Fig. S11). We choose a zonally uniform volcanic

cooling, but consider different meridional profiles, which emulate volcanic eruptions at different latitudes. Further details are given in Sec. 3.7.

## 3 Results

### 3.1 Time lags of DO event onsets to preceding volcanic eruptions

We estimated the precise times of the DO warming onsets from individual high-resolution Greenland $\delta^{18}$O ice core records, as well as from a stack derived from these. Due to the improved signal-to-noise ratio, the stacked record allows for the most precise onset determination, with an average uncertainty of the onset timings of roughly 13 years (Sec. 2.3). Thus, we will focus on discussing the results derived from the $\delta^{18}$O stack. Fig. 1 shows our estimates of the DO warming onset times and the ages of the $N = 82$ bipolar eruptions from SVE20 together with the stacked Greenland $\delta^{18}$O record. There are a considerable

number of DO events that are initiated within a short time after a volcanic eruption (see Fig. 1b-h). The time lags of the $M = 20$ DO warming onsets to the closest preceding bipolar volcanic eruptions are shown in Fig. 2a. For the following 7 interstadials the onset occurs within 50 years after a bipolar volcanic eruption: GI-1, GI-3, GI-4, GI-8, GI-9, GI-13 and GI-15.1. If we instead look at a tolerance of 20 years, we find 5 events: GI-1, GI-3, GI-8, GI-13 and GI-15.1.

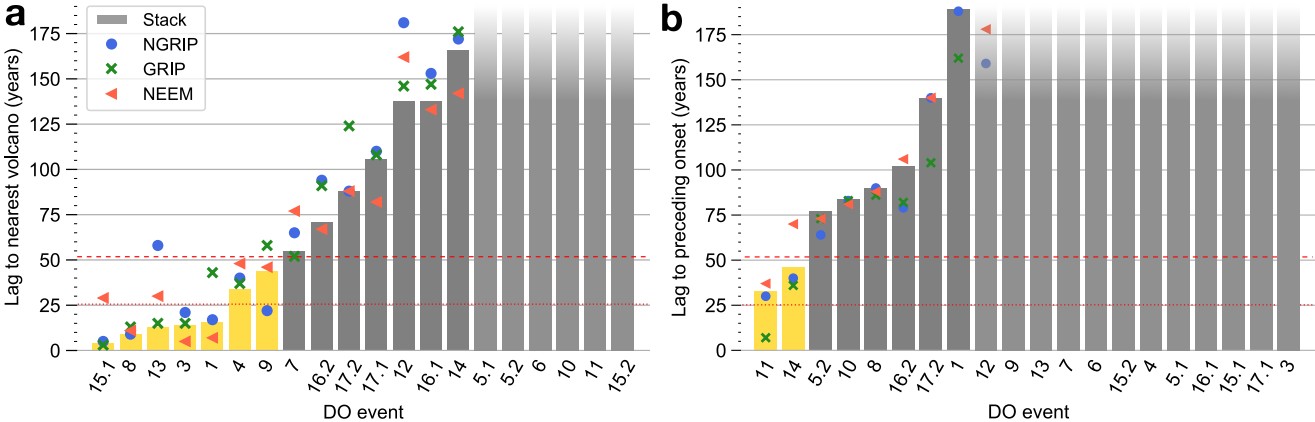

**Figure 2. a** Time lag of the DO onsets to the nearest preceding bipolar volcanic eruption. Bars and symbols indicate results for the isotope stack and individual ice cores, respectively. The yellow bars indicate events where the onset estimate in the stack occurs within 50 years after the nearest eruption. The red dotted and dashed lines mark the 5- and 10-percentiles of the time lags for a single event under the null hypothesis, respectively. **b** Same as **a**, but considering the nearest eruptions that occur after the DO onsets.

In Fig. 2b, the time lag of the closest eruption occurring after the DO warming onsets is shown. It is seen that these time lags

are much larger compared to the eruptions preceding the DO onsets. This indicates that the identification of bipolar eruptions shortly before DO onsets does not just reflect the fact that the ice cores are already well-synchronized close to the DO warmings compared to elsewhere in the record, potentially making it easier to find a bipolar match with the required confidence. If this

were the case and there were no causal relationship in between bipolar eruptions and the DO warmings, one would expect a roughly equal distribution of time lags for eruptions occurring before and after the DO onsets. The time lags for eruptions after the DO onsets may even be expected to be shorter, since there the bipolar eruptions should be easier to identify because of the higher interstadial accumulation rate and thus proxy resolution.

## 3.2 Magnitude of eruptions and upper bound on occurrence rate

To see whether this is significant, we need to estimate the number of eruptions that would occur shortly prior to DO onsets by chance if DO events and bipolar eruptions were uncorrelated. The frequency of occurrence of the large eruptions from SVE20 is approximately once every 500 years (see Sec. 2.4). To ensure that this is not a significant underestimate, due to a large number of eruptions potentially missing from the data set, we show that the bipolar eruptions were really of a magnitude that matches their occurrence frequency of once every 500 years using ice core records covering historic eruptions as well as a continuous data set of glacial volcanic eruptions by LIN22. Further, we derive an upper estimate on the number of equally large eruptions that could have potentially occurred and are not identified in SVE20. This upper estimate is then later used to test the robustness of our results to an underestimate of the occurrence rate.

While due to the deposition of significant amounts of sulfate at both poles it is evident that all events in the SVE20 data set were large eruptions, their magnitude may vary significantly. Recently, Lin and co-workers compiled a record of volcanic eruptions from continuous sulfate measurements in ice cores from Greenland and Antarctica in the interval 60 - 9 ka (see Sec. 2.2), and calculated their total sulfate deposition (LIN22). In their study, Lin and co-workers also quantified the bipolar eruptions from SVE20 as a subset. Thus, we can use the LIN22 data sets to compare the magnitude of the bipolar eruptions to all other eruptions in the same time interval, as well as in the last 2,500 years (Sigl et al., 2015), that are detectable in the ice cores. As detailed in Sec. 2.4, the magnitude of the 5 largest eruptions in the last 2,500 years matches the magnitude of the bipolar eruptions from SVE20, lending support to the observed frequency of once every 500 years. To see whether there could still be eruptions of similar magnitude that were not identified as bipolar, in the following the sulfate deposition of the bipolar eruptions is compared to the remaining 740 (498) eruptions detected in Greenland (Antarctic) ice cores (called *unipolar* hereafter), which could not be identified as bipolar eruptions in SVE20.

At both poles, the distribution of sulfate depositions corresponding to bipolar eruptions, as shown in the histograms of Fig. 3a-b, is clearly skewed towards larger values compared to the remaining (unipolar) eruptions. Within the population of bipolar eruptions, the 7 eruptions occurring within 50 years prior to a DO onset have a slightly smaller deposition on average, but this is not statistically significant. For Greenland, the average deposition of the events before DO onsets is 107.9 kg/km$^2$, compared to 157.3 kg/km$^2$ for the entire bipolar population. A bootstrap hypothesis test for the average shows that this difference in mean is not significant ($p = 0.16$). For Antarctica, there is an average of 51.1 kg/km$^2$, compared to 63.0 kg/km$^2$ for the bipolar population ($p = 0.28$). From the relative depositions in Greenland and Antarctica, LIN22 also derived the total stratospheric aerosol loading of the bipolar eruptions, which determines their global climate forcing. The distribution of this aerosol loading is shown in Fig. 3c. The mean for the eruptions before DO onsets (126.2 Tg) is again slightly lower than the population mean (159.9 Tg), but not significantly ($p = 0.21$). Thus, in terms of their magnitude the eruptions before DO

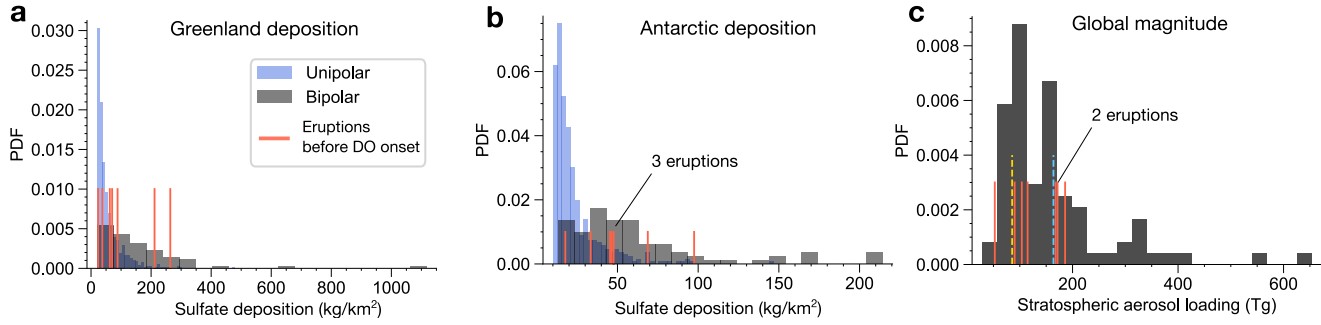

**Figure 3.** Histograms of the magnitude of volcanic eruptions in the last glacial period according to the LIN22 data sets. **a** The Greenland sulfate deposition of the eruptions is shown for the subset of eruptions classified as bipolar by SVE20, as well as the remaining eruptions identified in the Greenland ice cores during the periods 11.7-16.5 ka and 24.5-60 ka, labeled *unipolar* hereafter. The deposition level of the 7 eruptions that occur within 50 years prior to DO events is given by the red bars. **b** Same for the Antarctic sulfate deposition. **c** Estimated stratospheric aerosol loading of the bipolar eruptions. The yellow and blue dashed lines corresponds to the aerosol loading of the 1815 AD Tambora and 1257 AD Samalas eruptions, respectively (Gao et al., 2007; Sigl et al., 2015).

onsets can be considered a representative sample of the bipolar population, and there is no evidence for a bias of systematically matching smaller bipolar eruptions near the DO onsets.

While the eruptions in the unipolar data sets show clearly lower average deposition levels in both Greenland and Antarctica compared to the bipolar eruptions, there remain subsets of eruptions in the unipolar data sets with deposition levels that fit the
bipolar population. It is likely that some of these are in fact bipolar, even though this has not been possible to establish with the current ice core records and chronologies. We thus derive a conservative upper bound estimate on the number of bipolar eruptions of the same characteristic magnitude that may have been unidentified. Roughly speaking, the maximum number of eruptions in the unipolar data set of comparable sulfate deposition to the bipolar population should be identified, which can be done in several ways. A simple estimate is obtained by finding the subset of largest eruptions in either of the unipolar data sets
that has a mean deposition equal to that of the 7 eruptions that occur within 50 years prior to DO events (the DO population hereafter). From the Antarctic data set with 498 eruptions in total we find that the subset of the 84 largest eruptions has a mean deposition corresponding to the mean of the DO population. For the larger Greenland data set (740 eruptions in total) this subset is significantly larger, with the 212 largest eruptions having a mean deposition corresponding to the mean of the DO population.

In order to get a single fair estimate of the number of potentially unidentified large bipolar eruptions, one actually needs to pick the lower number of the two. This is because for a bipolar eruption of the size considered in our study, a sufficiently large deposition is required at both poles, and thus the subset needs to satisfy the required mean deposition in both poles in order to be comparable to the bipolar eruptions from SVE20. If one would consider the 212 largest eruptions in the Antarctic data set, one would obtain a subset with a mean deposition that is much smaller than the average of the DO population. Thus, there
cannot be this many additional eruptions in the ice core record that have a characteristic magnitude comparable to the data from

SVE20. Instead, the discrepancy in between the Greenland and Antarctic data sets implies that the Greenland data set contains a large number of regional (e.g. Icelandic) eruptions with a large Greenlandic deposition due to the proximity of the source, but a very small or absent Antarctic deposition, and thus a global impact that is not comparable to the bipolar eruptions from SVE20. This is supported by an analysis of volcanism in ice cores of the last 2,000 years compiled by Sigl et al. (2013), where

Greenland and Antarctic ice cores are very well synchronized throughout and the volcanic signals are preserved very well due to the large layer thickness. This leads to a very small number of potentially unidentified bipolar eruptions. Considering the 4 eruptions with the largest Greenland sulfate deposition during this time period, 3 eruptions do not have any associated Antarctic deposition, and are thus not classified as bipolar. 2 of them are of known Icelandic origin, with the 1783 AD Laki eruption being the largest of all. The remaining eruption of the 4 has an Antarctic deposition of less than 10kg/km$^2$, which

is the threshold that was used in the LIN22 data set. These 4 eruptions happen within 1,200 years, indicating a frequency of large Greenlandic unipolar eruptions that is even higher compared to the bipolar eruptions in the SVE20 data set. Note that this does not mean that such eruptions had no climatic impact. For instance, the climatic impact of the 1783 AD Laki eruption is still actively investigated (Zambri et al., 2019; Edwards et al., 2021), which is a difficult task for individual eruptions due to natural climate variability. But at least a pronounced global volcanic cooling seems less likely in the absence of Antarctic

sulfur deposition, and in the case of the Laki eruption it has been argued that the large Greenland sulfate deposition is largely a result of transport in the troposphere or lower stratosphere (Lanciki et al., 2012). In contrast to the large Greenland unipolar eruptions, the 4 eruptions with the largest Antarctic sulfur deposition, which include the 1815 AD Tambora and 1257 AD Samalas eruptions, all have a large associated Greenland deposition.

This further supports the usage of the Antarctic data set to constrain the number of potentially missing bipolar eruptions,

in addition to the abovementioned methodological requirement of a large deposition at both poles. As a result, following this methodology we find an upper bound estimate of 84 potentially unidentified bipolar eruptions. Based on this estimate, the bipolar SVE20 data set used in our study could be missing 50% of the eruptions of similar magnitude at maximum. We call this an upper bound, because, while seemingly less pronounced compared to the Greenland data set, also the Antarctic data set contains regional eruptions with a strong Antarctic deposition, but a weak or absent Greenland deposition and global climate

impact. This is partly due to local Antarctic volcanism, which can be detected in Antarctic ice cores throughout the last glacial (Narcisi et al., 2017).

The estimate is furthermore conservative, because the distribution of the sulfate deposition of this potential subset of 84 eruptions is strongly skewed. This is because the tail of the unipolar distribution decays much faster than the bipolar distribution (see Fig. 3b). Smaller deposition events are thus represented more frequently than in the bipolar population. Since our analysis

is based solely on a population of events within a given size distribution, it may be more fair to ask the following: What is the maximum number of bipolar events that the unipolar sample could contain and that follow the size distribution of the known bipolar population? Since such a subset of events will mainly come from the tail of the unipolar distribution, and the events in the tail are relatively sparse, a probabilistic approach is adopted in order to obtain an uncertainty in the estimate of the upper limit. The method is detailed in the Appendix B, and uses random sampling with replacement to repeatedly

generate maximal subsets from the unipolar population, which follow the bipolar deposition distribution. Fig. 4a shows that

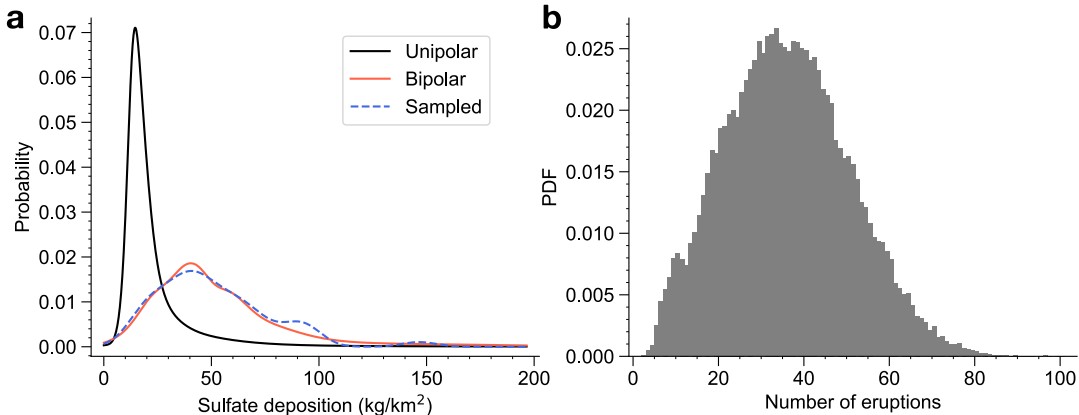

**Figure 4. a** Kernel density estimate of the Antarctic sulfate deposition in the unipolar and bipolar LIN22 data sets, as well of a large number of samples from the resampling method described in Appendix B (dashed blue line), where subsamples from the unipolar data set are drawn such that they follow the distribution of the bipolar population. **b** Histogram of the number of eruptions contained in each unipolar subsample after 50,000 iterations of the resampling method.

these sampled subsets indeed follow this distribution to a good approximation. In Fig. 4b we show the distribution of the number of events in the subsets. The mean number of events in the Antarctic data set is 37.5, with a confidence interval of [20.0, 55.0], based on the 10- and 90-percentiles. Interestingly, using the Greenland data set instead, we find a slightly smaller number of 32.8 eruptions on average (see Fig. S7), as opposed to the much larger number of eruptions with a mean deposition of the DO population reported above. This is because the distribution of Greenland deposition in the bipolar data set is much more skewed compared to the Antarctic data set (compare Fig. 4a and Fig. S7a). Thus, the mean is dominated by the largest eruptions and a subset constructed based on the mean will contain many more small eruptions to balance out the large ones. We thus find that, as expected, taking into account the observed distribution of the deposition corresponding to bipolar events instead of only matching the mean deposition leads to a smaller estimate. We will, however, use the larger upper bound of 84 missing eruptions to test the robustness of our results.

The somewhat naive approach of using the unipolar depositions to infer the magnitude of potential bipolar eruptions can ultimately not reveal the true frequency of eruptions with bipolar imprint and large global climate forcing. Thus, we want to stress that the numbers given above are not our best estimates on the actual number of unidentified bipolar eruptions, but rather an upper bound of such, where we assume that most of the large deposition events at the individual poles are indeed eruptions with a bipolar imprint. Further, the numbers given here, i.e., the return period of 500 years as well as the estimate of eruptions potentially missing from the SVE20 data set, do not refer to bipolar eruptions of any size, but to bipolar eruptions of the characteristic (large) size of the bipolar eruptions in the SVE20 data. The former are indeed known to occur much more frequently (Sigl et al., 2022). The actual frequency of large bipolar eruptions remains an open question until new data becomes available. For our purpose of constraining the maximum frequency of events from within the currently available data sets, the estimates given here are sufficient.

### 3.3 Comparison of data to null hypothesis of randomly occurring volcanic eruptions

Given the number of eruptions close-by to DO onsets found in Sec. 3.1, we now ask whether this is likely to be observed by chance in the case that bipolar volcanic eruptions occurred completely randomly and uncorrelated to the occurrence of DO events. To answer this, we test the null hypothesis that volcanic eruptions occur as a stationary Poisson process independently of DO events, i.e., they do not preferentially occur shortly before DO onsets. The description of volcanic eruptions as stationary Poisson process means that they happen at a constant rate $\lambda = N \cdot (\Delta T)^{-1}$ over time and that subsequent eruptions are not correlated (De la Cruz-Reyna, 1991). Our eruption data with $N = 82$ and $\Delta T = 40.3$ kyr are consistent with this assumption, yielding $\lambda = 2.03$ eruptions per kyr (Sec. 2.4). The waiting times in between subsequent DO onsets can also be assumed to be uncorrelated. A two-tailed bootstrap hypothesis test on the Pearson (Spearman) correlation of consecutive waiting times yields $p = 0.60$ ($p = 0.32$) for a correlation of $r = 0.13$ ($r = 0.26$). For this interval of the glacial, which includes MIS-3 with rather regular DO cycles, the empirical distribution of waiting times does not fit perfectly to an exponential function. An Anderson-Darling (Kolmogorov-Smirnov) test almost rejects the exponential distribution at 95% confidence with $p = 0.060$ ($p = 0.051$). However, the resulting potential deviation of the DO onsets from a Poisson process is not important since the average spacing in between the onsets is large compared to the tolerance window and spacing of eruptions. Thus, the data set is suited for our null hypothesis.

Under the null hypothesis, the expected value for the time lag of an independently occurring DO event to the closest preceding volcano is $\lambda^{-1} = 491$ years. The quantiles for the time lag are $-\lambda^{-1}\ln(1-p)$. Thus, there is a 5% (10%) chance of observing a time lag smaller or equal to 25.2 (51.8) years. The horizontal lines in Fig. 2a show that in both cases there is a considerable number of events with smaller time lag. To show that this is indeed significant, we say in the following that there is an event match, if a volcanic eruption occurs within a time lag window of $\tau$ years prior to the DO onset, which we call the tolerance. The number of event matches in the data for tolerance $\tau$ shall be compared to the number of times one or more volcanic events would be found in a time window of $\tau$ years by chance when randomly sampling $M$ windows of a Poisson process with rate $\lambda$. In this case the number of events $n$ occurring during $\tau$ years is given by the Poisson distribution. The probability to observe one or more events within $\tau$ years is

$$P(n(\tau) \geq 1) = 1 - e^{-\lambda\tau}. \tag{2}$$

Randomly choosing $M$ independent windows and observing whether they contain any events is equivalent to a sequence of Bernoulli trials with success probability $P(n(\tau) \geq 1)$. Thus, the probability of finding one or more events in $k$ out of $M$ windows is given by the Binomial distribution:

$$P_\tau(k) = \frac{M!}{(M-k)!k!}(1 - e^{-\lambda\tau})^k e^{-\lambda\tau(M-k)}. \tag{3}$$

Finally, the probability of finding at least $k$ out of $M$ windows containing one or more events represents the $p$-value for our null hypothesis. This is the cumulative probability of the previous expression

$$p \equiv P_\tau(K \geq k) = 1 - \sum_{l=0}^{k-1} \frac{M!}{(M-l)!l!}(1 - e^{-\lambda\tau})^l e^{-\lambda\tau(M-l)}. \tag{4}$$

For the $k = 7$ matches we find at $\tau = 50$ years, this yields a probability of $p = 0.002$. Thus, we can reject the null hypothesis at a confidence level of 99%. For the $k = 5$ matches found at $\tau = 20$ years we obtain $p = 0.0009$. The expected value of matches is $E[k] = M(1 - e^{-\lambda \tau})$, yielding only 1.9 (0.8) events preceded by one or more eruptions within 50 (20) years under the null hypothesis. In contrast, the time lags of eruptions occurring after the DO onsets (Fig. 2b) correspond well to this null hypothesis.

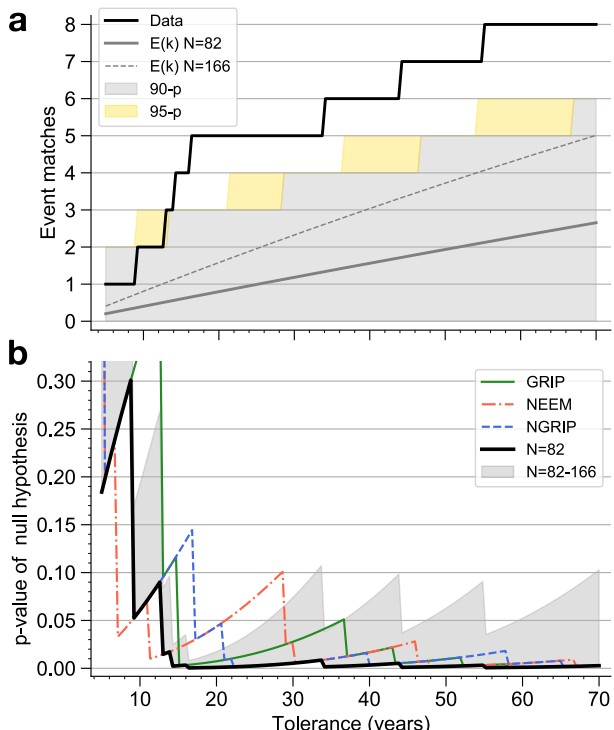

**Figure 5.** Probabilities of the observed number of DO events occurring shortly after volcanic eruptions under the null hypothesis. **a** Number of event matches for the stack onsets (Data) as a function of the tolerance, compared to the expected value $E[k]$ as well as 90% (90-p) and 95% (95-p) confidence bands of the null hypothesis. Here, the upper limit of the bands correspond to the smallest number of matches that have a probability of strictly less than 10% (5%) according to Eq. 4. **b** Probability to observe at least as many event matches as in the data under the null hypothesis, calculated from Eq. 4. These probabilities are the p-values of the null hypothesis. The thick black line shows the results for the stack onsets and the Poisson null hypothesis using $\lambda = N/\Delta T$ with $N = 82$, being the number of large eruptions in the investigated time interval. The gray shading indicates the range of probabilities when doubling $N$ from $N = 82$ to $N = 164$. Results for the onsets from individual cores with $N = 82$ are also shown.

## 3.4 Robustness of results to varying choices of maximum time lag

Our confidence in rejecting the null hypothesis may depend on the tolerance $\tau$. To see whether the results are robust, we consider the binomial probability for a plausible range of tolerances. Tolerances lower than 10-15 years should not be considered

because they are of the same order as the estimated uncertainty in our DO onset timings. Furthermore, at these tolerances event matches in the data become too rare to be a reliable estimate of the actual probability of co-occurrences of DO events and volcanoes. For tolerances of a century or more, a direct climatic influence of even large volcanic eruptions becomes less likely. Figure 5a shows the number of matches as a function of $\tau$. The solid gray line indicates the expected value of matches

$E[k] = M(1 - e^{-\lambda\tau})$ under the null hypothesis. The gray (yellow) shading indicates the corresponding 90% (95%) confidence bands. For tolerances larger than 13 years, the data lie consistently above the 95% confidence band, and thus are unlikely to occur under the null hypothesis. The precise probabilities as calculated from Eq. 4 are given in Fig. 5b. These correspond to the p-values of the null hypothesis as a function of $\tau$, and we see that the data lie below $p = 0.01$ for all tolerances larger than 13 years. The results are also significant when considering the onset timings of the individual ice cores (colored curves).

Further, the analytical results on the probability of event matches occurring under the null hypothesis reported in this Section and the previous one are accurate and robust to the (minor) deviations of our data sets from the assumptions underlying the null hypothesis. This is shown in Fig. S6, where the analytical results are compared to direct Monte Carlo simulations where the actual observed DO onset and volcanic eruption data is used explicitly.

## 3.5   Robustness of results to a potential underrepresentation of bipolar eruptions

Most importantly, the confidence at which we can reject the null hypothesis depends on the estimate of the occurrence rate of eruptions. If the SVE20 data set is incomplete, the frequency of eruptions $\lambda$ is underestimated. This has been discussed in detail in Sec. 3.2, and we found 84 eruptions as a conservative upper bound estimate of the maximum number of potentially unidentified eruptions. When increasing $N$ from 82 to 166, the expectation value $E[k] = M(1 - e^{-\lambda\tau})$ almost doubles, as can be seen by the dashed line in Fig. 5a. Still, the number of matches in the data remains significant: The shading in Fig. 5b shows

the range of probabilities obtained when considering rates $\lambda = N \cdot (\Delta T)^{-1}$ by increasing $N$ from 82 to 166. Even then we consistently find $p \leq 0.1$, allowing us to reject the null hypothesis at 90% confidence. Thus, our results are robust to a potential under-counting of large bipolar eruptions where half of the eruptions not occurring close to a DO event had not been identified.

## 3.6   Contrasting volcanic influence on warming and cooling phases of DO cycles

It may seem counter intuitive that the radiative aerosol cooling of volcanic eruptions would induce abrupt warming transitions

of the DO cycles. An effect on the abrupt cooling phases of DO cycles may seem more likely at first. We performed the same analysis on the cooling transitions that mark the terminations of interstadials and do not find a significant clustering of eruptions leading up to the coolings (see Fig. S5). We note, however, that the cooling transitions in the isotopic records are generally much less well-defined. The statistical basis to assess the significance of this is thus not as solid compared to the case of the DO warmings. As a result, the question of a potential volcanic trigger of DO cooling transitions remains inconclusive, as discussed

in the following.

Firstly, about half of the interstadials do not have a well-defined abrupt cooling, and one might thus argue that it is not meaningful to look for a trigger of the terminations of these interstadials. In Lohmann and Ditlevsen (2019) criteria are given and tested for the saw-tooth shape of DO cycles, i.e., the existence of an abrupt cooling that is distinct from the gradual cooling

during the interstadials. Out of the 19 interstadials investigated here, 10 do not show a distinct abrupt cooling according to Lohmann and Ditlevsen (2019). When restricting the sample to the 9 events with a clear abrupt cooling, the closest distance of a preceding volcanic eruption is 170 years. Considering that the 10-percentile remains 51.8 years (Sec. 3.3), this is clearly not statistically significant. Out of all 19 cooling transitions, we find only one DO cooling where a preceding eruption occurs within 100 years. Based on these timing estimates, it seems impossible to find a subset of events where a statistically significant link of interstadial terminations and volcanic eruptions can be established.

However, secondly, there are larger uncertainties in our estimated transition timings, determined here by a different method introduced in Lohmann and Ditlevsen (2019), as compared to the abrupt warmings. This is evident by the relatively large inter-core spread in Fig. S5a, and it holds also for the interstadials with a pronounced abrupt cooling. As a result, we might have missed individual event matches, such as the volcanic eruption occurring close to the termination of GI-16.2. The short inter-stadials GI-16.2 and GI-17.2 (where we do find a volcanic eruption closely preceding the termination) are outliers (Lohmann and Ditlevsen, 2019). Thus, a volcanic influence to end them prematurely might be plausible. Thirdly, some large bipolar events close to the coolings could have been missed by the SVE20 data, given that the prior synchronization of Greenland and Antarctic ice cores is not as good close to DO coolings compared to the DO warmings.

Keeping in mind these uncertainties in the existence and timing of the abrupt coolings, the current bipolar data suggests that such volcanic influences before DO coolings did not occur more frequently than would be expected by chance. However, in our study we do not consider the potential impact of large extra-tropical eruptions without a bipolar imprint, which can also exert a significant long-lasting impact on the NH climate (Kobashi et al., 2017; Toohey et al., 2019; van Dijk et al., 2022). To explain the influence of volcanic cooling on DO warming or cooling transitions, one needs to consider its impact on the major parts of the climate system that are believed to be involved in DO cycles. In the following, we explore a very simple mechanism that considers the regional cooling impact on the AMOC, using a global ocean model.

### 3.7 Experiments with a global ocean model to explore the impact of volcanic cooling on DO cycles

Transitions in between a vigorous and a collapsed or very weak circulation state of the AMOC are commonly implied to explain the known spatio-temporal patterns of DO cycles. Whether the transitions occurred as autonomous, self-sustained cycles, as a result of gradual changes in climate background conditions, or at least partly by stochastic forcing is currently not known. Regardless of the long-term dynamics, these mechanisms have in common that due to non-linearity the alternative stable AMOC states co-exist for a range of climatic background conditions. Transitions can occur either when the background conditions change so much that one of the states loses stability (at a tipping point), or when the system is perturbed quickly by a sufficiently large shock. The latter becomes more likely the closer to a tipping point.

The strength of the AMOC depends on the convection in the NA, where dense water is formed at the surface, sinks towards the bottom and then flows southwards. The NA surface density is influenced by atmospheric temperature, as well as the salinity budget via freshwater exchange. A weakening and eventual collapse of the AMOC can occur if too much freshwater enters the NA, e.g., from Greenland melt. In Fig. 6a we show the response of the AMOC strength in a global ocean model *Veros*, where a freshwater forcing is introduced to the NA, slowly increased, and finally decreased again (see Sec. 2.5 for more details).

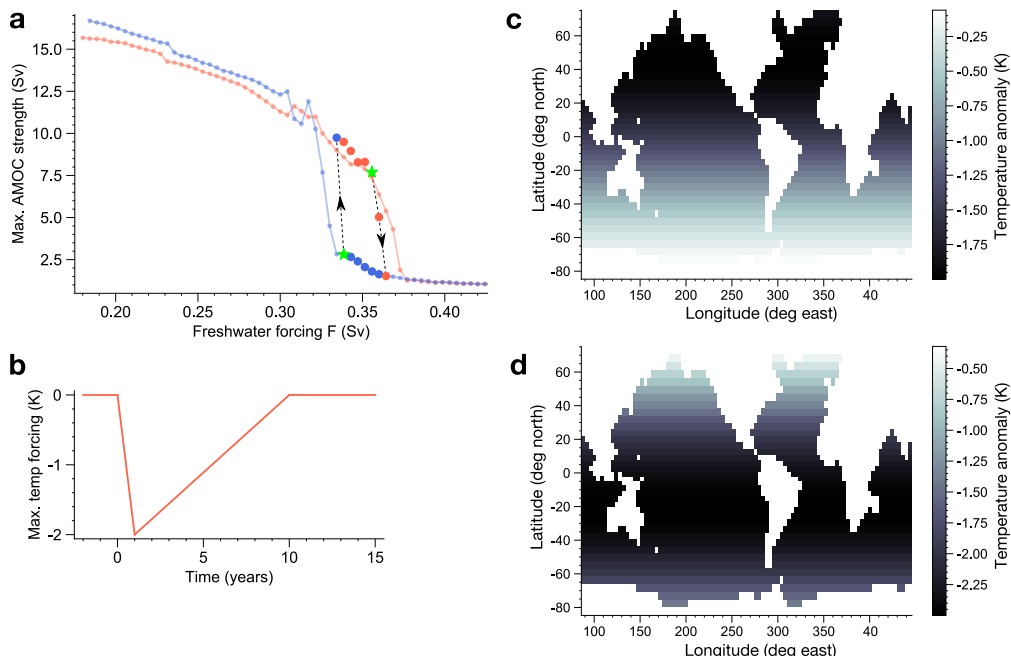

**Figure 6. a** Hysteresis diagram for the maximum AMOC strength below 500 meters depth in the *Veros* model as a function of the NA freshwater forcing $F$. The small circles show the model states during the transient hysteresis experiment, where red (blue) symbols are the results for increasing (decreasing) $F$. The large circles represent the model states after long equilibrium simulations at fixed $F$, which were branched off from the hysteresis experiment (see Fig. S10 for the corresponding timeseries). In the lower branch (collapsed AMOC, blue circles), the circulation resurges when decreasing the freshwater forcing to $F = 0.334$. When increasing the freshwater forcing in the upper branch (red circles), the circulation collapses first partially at $F = 0.360$, and then fully at $F = 0.364$. The green stars represent the last stable states observed before the tipping points. **b** Temporal evolution of the volcanic cooling forcing. **c-d** Spatial pattern of the temperature forcing anomaly for a NH/equatorial eruption (**c**) and a SH eruption (**d**). Shown are the maximum values of the cooling one year after the eruption. For the remaining times, the cooling is scaled uniformly according to the temporal profile in **b**.

This reveals a regime of the freshwater forcing parameter $F$ with co-existing vigorous and collapsed AMOC states, which are analogous to the conditions of the AMOC during the last glacial period: Stadial periods were generally associated with a collapsed AMOC, while in interstadials a vigorous AMOC was found (Henry et al., 2016). The stability of the model states is confirmed with long equilibrium simulations (large circles in Fig. 6a). See Fig. S9 for an illustration of the corresponding
5  AMOC stream functions for the collapsed and vigorous circulation state at the same forcing $F$, and Fig. S10 for time series of the maximum AMOC strength of the equilibrium simulations. The range of bistability (roughly 0.02 Sv) is quite narrow compared to other ocean-only models and climate models of intermediate complexity, and unlike in many other models the present-day climate is not in the bistable regime (Rahmstorf et al., 2005). However, the bistability range in this ocean-only framework is dependent on parameters that are largely unconstrained, such as the temperature and salinity relaxation time

scales (Lohmann and Ditlevsen, 2021). The width and location of the bistability regime does not crucially influence the ability to test our hypothesis.

Given that two alternative stable states exist, the direct thermal influence of volcanic cooling on the AMOC might trigger a transition from a collapsed to a vigorous AMOC (a DO warming onset), but not vice versa. Specifically, NA atmospheric cooling leads to increased heat loss of the ocean and thus dense water formation, which can initiate local convection. Thus, a spontaneous resurgence of a collapsed AMOC may be induced, while conversely a spontaneous collapse of a vigorous AMOC becomes less likely. Since the perturbation is relatively short-lived, a spontaneous transition would only be likely if the system is already close to a tipping point.

We test this in the model with two sets of simulations, initialized in the vigorous and collapsed AMOC state, respectively. The background conditions in terms of the NA freshwater forcing are chosen such that the system is close to a tipping point in both scenarios. The corresponding states are marked with a green star in Fig. 6a. Simulations started in the partially collapsed state at $F = 0.360$ have also been performed (Fig. S13b). Starting from long spin-up simulations with constant boundary conditions, we introduce a global volcanic cooling that peaks one year after the eruption and gradually fades out within 10 years (see Fig. 6b). We consider two different spatial patterns, with more pronounced NH and SH cooling, respectively (see Fig. 6c-d). The former scenario could represent NH and equatorial eruptions, which have both shown more pronounced NH cooling in model simulations (Schneider et al., 2009; Pausata et al., 2020; Black et al., 2021; Zhuo et al., 2021). The scenario with stronger SH cooling would then rather correspond to an extra-tropical SH eruption.

Figure 7 shows that a transition of the AMOC can only be observed in the case of the NH/equatorial eruption starting from the collapsed state (green trajectory in Fig. 7a). This is robust under changes of eruption season, magnitude and initial conditions (Fig.'s S9 and S10). The SH eruption does not lead to an AMOC resurgence, plausibly due to insufficient NA cooling. Both simulations initialized in the vigorous AMOC state show an abrupt strengthening of the AMOC, followed by damped oscillations back to the original state. We note that the transition in Fig. 7a is not very abrupt, as opposed to the observed DO onsets. While as a result of the onset of NA deep water formation the maximum AMOC strength initially increases abruptly after the volcanic perturbation, the initial strong NA convection is replaced by a more gradual widening of the overturning cell after a couple of decades. In dynamical systems terms, the initial perturbation of the AMOC pushes the system from the collapsed stable state not directly into the vigorous stable state, but into its basin of attraction. The transition within this basin of attraction is then very gradual and may exhibit transient slowing down of the dynamics as the vicinity of one or more saddle states is visited, which we believe is specific to our model and its configuration (e.g. the configuration in Lohmann and Ditlevsen (2021) behaved much more abrupt). In physical terms, the slow transition may also display the fact that the model does not include active components with fast time scales, i.e., dynamic sea ice and atmospheric dynamics. Especially sea ice dynamics are believed to be involved in the abruptness of DO events (Li et al., 2005; Dokken et al., 2013).

Still, we believe the model simulations help to illustrate a potential explanation of our observation that DO warming transitions seem to be associated with preceding eruptions, while data set used here we could not find similar evidence with respect to DO cooling transitions. The observation that only some DO onsets are preceded by eruptions is consistent with the idea

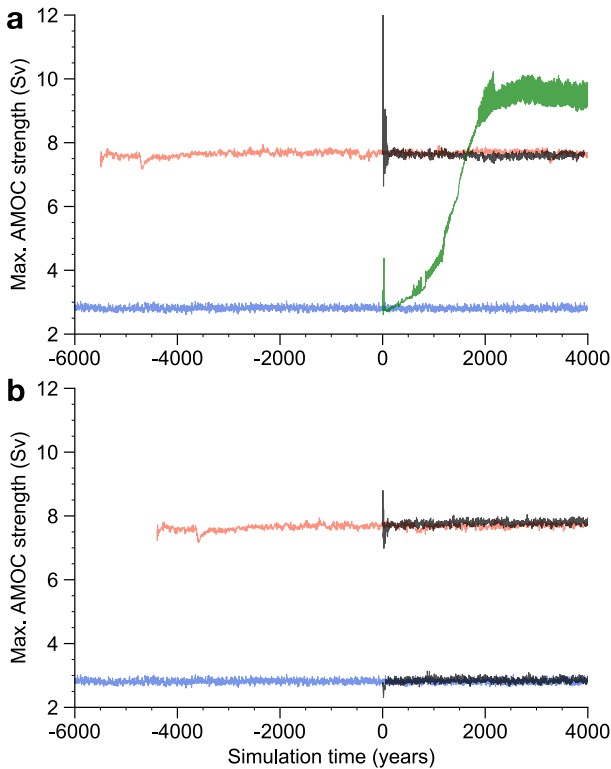

**Figure 7.** Time series of the maximum AMOC strength in ocean model simulations where a short-term volcanic cooling is incorporated into the atmospheric temperature boundary condition. The simulations (shown in black/green) are branched off in year 0 from the equilibrium simulations in the collapsed (blue) and vigorous (red) AMOC states, and the volcanic perturbation is started simultaneously. The simulations with the NH/equatorial and SH scenario are shown in panels **a** and **b**, respectively.

that an eruption needs to happen at a point in time when the climate system is already close to a tipping point, which is also supported by our model simulations (see Fig. S13a for simulations initialized further away from the tipping point).

It is well-known that the latitude and Hemisphere of an eruption influences its climatic impact (Sun et al., 2019; Zhuo et al., 2021). If indeed a large NA cooling is required to trigger a DO warming, one might argue that not all of the bipolar eruptions considered here should be included in our analysis. LIN22 classified the bipolar eruptions into two latitude bands by considering the relative deposition in Greenland versus Antarctica, with the following results. Out of the 82 bipolar eruptions, 34 are classified as low latitude or Southern Hemisphere (Low Latitude below $40^o$N or Southern Hemisphere, LLSH), while 48 are classified as Northern Hemisphere High Latitude (above $40^o$N, NHHL). Regarding the eruptions identified here as occurring shortly before DO onsets, the classification yields 5 (LLSH) vs 2 (NHHL) eruptions occurring within 50 years, and 4 (LLSH) vs 1 (NHHL) eruptions within 20 years. This may seem to be conflicting our proposed trigger mechanism via NA cooling. However, note that the LLSH classification includes all tropical eruptions, which are still expected to have a large NH cooling, as discussed briefly above. Only few large extra-tropical SH eruptions are known during the last glacial (LIN22).

In general, while the classification by LIN22 is helpful, we do not believe it warrants us to a priori favor certain eruptions in our statistical analysis. There is a large uncertainty in the classification of individual eruptions due to the large inter-core variability of the sulfate deposition. Further, the relative sulfate deposition does not fully constrain the actual climate impact of the eruptions, which also depends on where and how long the aerosols were transported. Still, the classification by LIN22 in relation to our results should be kept in mind for future investigations.

## 4   Discussion and Conclusions

Here we present evidence that initiations of DO warming events within a short time after large volcanic eruptions happened more frequently than can be expected by chance. Whereas previous research showed that DO event onsets are predictable in principle (Lohmann, 2019), there could be additional factors that influenced the timing of some abrupt warmings, such as the perturbations invoked by the volcanic eruptions identified here. These could be interpreted as short-term, stochastic triggers influencing the timing of DO onsets, which, in the absence of such triggers, might eventually occur regardless due to processes that happen on longer time scales. Indeed, for 3 out of 4 events where we find an eruption within 20 years prior to the onset, it occurred earlier than predicted by Lohmann (2019) (GI-1 was not part of that study).

Using a new record of bipolar volcanic eruptions in conjunction with multiple high-resolution Greenland ice core records, it is for the first time possible to perform the analysis presented here with the required temporal precision, so that a direct climatic influence of eruptions on the abrupt climate transitions is plausible without having to invoke centuries-long climate feedbacks. When determining significance of the results, our method uses tolerance windows and is thus not sensitive to errors in the timing of the onsets on a sub-decadal time scale. This includes small potential offsets in the initiations of DO warmings when estimated from other proxies (Erhardt et al., 2019; Capron et al., 2021; Riechers and Boers, 2021).

As with most other multi-proxy statistical analyses, a drawback of our study is that we cannot infer directly whether there is a true causal connection from eruptions towards the climate (DO onsets). Alternatively, there may be confounding factors that influence both the occurrence of DO events and volcanic eruptions, or there may be a direct feedback of the climate state onto the volcanic activity that is dominant compared to the influence of eruptions on the climate. The most likely process for this is the changing mantle stress due to melt and growth of glaciers and ice sheets, which can modulate volcanic activity in time (Cooper et al., 2018). While on a global scale this modulation is mostly associated with glacial-interglacial cycles, it could be relevant on multi-centennial time scales on a regional level (Swindles et al., 2018). Still, we do not believe that our data set is significantly modulated in this way, since the eruption rate in stadials and interstadials is the same, and the data is consistent with a stationary process (Fig.'s S3 and S4). Further, we test against an undercounting of eruptions, which means that our results are robust towards a certain degree of potential modulation that would generally favor eruptions around stadials or at the end thereof. Nevertheless, more studies are needed to rule out this hypothesis and other confounding factors. For a meaningful statistical analysis on the dependence of volcanic activity on mean global climate or global ice volume, an even longer record of volcanism (covering the whole glacial) is crucial, and we are currently working on this. At the same time, better data on potential confounding factors is needed, such as well-dated ice volume records that resolve DO cycles.

A remaining challenge of our analysis is that the present data set of glacial bipolar eruptions has not been obtained with automated methods. Thus, it may be missing unidentified bipolar eruptions of similar magnitude, or have a systematic bias favoring the identification of eruptions close to abrupt transitions, both of which would lead to an underestimate of the occurrence frequency and of the number of eruptions that are expected to happen close to DO onsets by chance. However, using a recently

published data set that objectively quantifies the magnitude of eruptions in Greenland and Antarctic ice cores during the last glacial (LIN22), including the bipolar eruptions considered here, we could address this issue. First, most bipolar eruptions during the glacial in the SVE20 data set have an estimated magnitude larger than the 1815 AD Tambora eruption, which itself is estimated as the sixth largest eruption of the last 2,500 years (Sigl et al., 2015). Thus, the observed occurrence frequency of one eruption per 500 years in our bipolar data set is quite consistent with the arguably more accurate observations of the

recent 2,500 years (Sec. 2.4). Assuming the frequency of large eruptions was not very different in the glacial, this indicates that there should not be many eruptions missing in our data set. Second, a comparison to the deposition levels of the eruptions in the continuous LIN22 data set, which were not identified as bipolar, allowed us to give an upper limit estimate on the number of potentially missing bipolar eruptions of similar magnitude. Even with a conservative estimate that essentially disregards the fact that some deposition events correspond to local eruptions of limited global impact, we find that our results remain

significant at 90% confidence. The potential bias of the SVE20 data set to a priori favor the identification of eruptions close to abrupt transitions - either by choice or because of the better prior synchronization of the records at the abrupt transitions - seems unlikely as the eruptions in question are not significantly smaller compared to the rest (Sec. 3.2), and since there is no clustering of eruptions occurring shortly after the abrupt transitions (Sec. 3.1). As a result, while the SVE20 bipolar volcanic catalogue certainly undercounts the true number of bipolar volcanic events of arbitrary strength, we argue that it captures a

sufficiently large portion of the strongest events most relevant to triggering climate change. While our analysis only considers bipolar volcanic eruptions that have been identified in the glacial sections of the ice cores used, volcanic events restricted to either the northern or southern hemisphere may likewise contribute to abrupt climate change. However, uncertainty in assessing their latitude and magnitude precludes us from evaluating them here.

As a conclusion, we find it very likely that the large volcanic eruptions that occurred a few decades before a significant

subset of the DO warming transitions contributed to the occurrence of the latter. This is consistent with previous studies which, unlike our study, featured multi-century timing uncertainties (Bay et al., 2004, 2006; Baldini et al., 2015), or allowed for very long time lags in between eruptions and DO transitions (Bay et al., 2004). In contrast to the abovementioned studies, the data analyzed here does not yield evidence for a similar statistical relationship of eruptions preceding the abrupt DO cooling transitions. This absence of evidence may, however, be a result of our restriction to eruptions with clear bipolar imprint, as

well as uncertainties in properly defining the abrupt cooling events (Sec. 3.6). The suggested triggering of abrupt warming events, on the other hand, by the global cooling of large volcanic eruptions may seem counter intuitive. However, one needs to keep in mind that the actual response of different parts of the climate system can be rather complex, and depend on both the site of the eruption as well as the season (Robock, 2000). The radiative aerosol cooling is typically not uniformly distributed over the globe, and tropical as well as high latitude eruptions can lead to a hemispherically asymmetric climate response

(Pausata et al., 2015; Black et al., 2021; Yang et al., 2022), and altered equator-pole temperature gradients (Pausata et al.,

2020). This asymmetric cooling leads to changes of the oceanic and atmospheric circulation, which could have impacted the glacial climate in many different ways. For instance, northward (southward) shifts of the inter-tropical convergence zone after SH (NH) volcanic eruptions were purported to be able to initiate abrupt DO warming (cooling) in the NH by Baldini et al. (2015), when amplified by further feedbacks likely associated with sea-ice extent in the NA.

Our finding of a potential volcanic trigger of DO warming transitions may be due to the direct thermal effect of volcanic cooling on the NA surface density, which in turn controls deep water formation and the AMOC strength. In the context of global warming, the decreasing heat loss and density of surface waters in NA convection regions leads to an AMOC decline in future climate projections (Gregory et al., 2005). Conversely, in a stadial climate state during the last glacial with collapsed AMOC, volcanic cooling could lead to increased ocean heat loss, deep water formation, the onset of NA convection, and

finally a resurgence of the AMOC to a vigorous state. Especially if the climate system is bi-stable and close to a tipping point, relatively small and short-lived perturbations of the AMOC could lead to a transition from stadial to interstadial conditions. We illustrated this concept with a global ocean model that features a bi-stable AMOC, and found that an eruption which leads to sufficient NA cooling can induce a transition from a collapsed to a vigorous AMOC state. A corresponding transition from vigorous to collapsed AMOC is not found, as the volcanic cooling has a strengthening influence on the AMOC in

this case as well. While this is consistent with our data analysis, the simulations with an ocean-only model omit important processes, and our simulations cannot reproduce the abruptness of DO events, despite the presence of tipping points. The missing feedbacks of the sea ice and atmosphere may alter the ocean heat loss after volcanic cooling and potentially even overrule the proposed ocean response to the NA cooling. Thus, the proposed mechanism needs to be tested with coupled atmosphere-ocean models under realistic glacial boundary conditions, where the atmospheric response is more nuanced and

includes changes in wind stress and precipitation patterns. While there are studies considering the impact of volcanic eruptions on the AMOC in comprehensive models, these are not in the context of DO events and the last glacial. Still, different models confirm a direct AMOC strengthening as a response to NH high-latitude (Pausata et al., 2015) as well as tropical eruptions (Stenchikov et al., 2009; Swingedouw et al., 2015; van Dijk et al., 2022; Liu et al., 2022), while more variable excitations of NA and AMOC variability have also been reported (Mignot et al., 2011), which could act as a source of noise to trigger

transitions of a bi-stable AMOC. More firm conclusions regarding the mechanism can only be achieved as additional data or model simulations become available. Nevertheless, the statistical link in between large volcanic eruptions and abrupt climate change shown here enhances our understanding of the causes of abrupt climate change, as well as of potential future impacts of large volcanic eruptions.

*Code and data availability.*  The bipolar volcanic record is available in the supplementary material of SVE20. The magnitude estimates of

volcanic eruptions from Greenland and Antarctica based on continuous sulfate measurements are available in the supplementary material of LIN22. The high-resolution NGRIP oxygen isotope record is available at http://iceandclimate.nbi.ku.dk/data/NGRIP_d18O_and_dust_5cm.xls. The GISP2 record is available at http://depts.washington.edu/qil/datasets/gisp2_main.html. The NEEM high-resolution oxygen isotope

record is available at https://doi.org/10.1594/PANGAEA.925552. The GRIP record is available upon request from the corresponding author. The source code of the Veros ocean model is available under a GPL license on GitHub (https://github.com/dionhaefner/veros).

*Competing interests.* The authors declare no competing interests.

*Acknowledgements.* We thank Roman Nuterman, Markus Jochum, Dion Häfner, and the Danish Center for Climate Computing for sup-
5    porting the simulations with the Veros ocean model. Further, we thank Reik Donner for helpful comments on a previous version of the manuscript. This project is TiPES contribution #68. The project has received funding from the European Union's Horizon 2020 research and innovation programme under grant agreement No. 820970.

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

**Appendix A: Method to determine the timing of DO event onsets**

This Section presents our method to determine the DO onsets from high-resolution $\delta^{18}$O records. We first divide the $\delta^{18}$O record roughly into DO cycles, to be used later individually for the onset determination. To do this, we use the piecewise-linear method from Lohmann and Ditlevsen (2019), which divides the record into DO cycles that comprise stadial and interstadial periods, as well as abrupt warming and cooling phases. While the breakpoint from stadial period to abrupt warming phase
obtained by this method already gives an estimate for the DO onset, it is not as precise as our following method that focuses on the DO onset only. This is because piecewise-linear methods find a compromise that fits all data points before, during and after the transition, and because noise blurs the sharpness of the transition, and the latter is not always linear. Thus, we use this method only to divide the record into DO cycles, and to get a rough estimate on where to look for the DO onset point. For this, other methods and approaches, such as using the transitions defined in Rasmussen et al. (2014), would work equally well.

Given a segment of the record that comprises a stadial period, the abrupt warming transition and the following interstadial period, the principle of our method is to identify a time point where the abrupt warming transition is already clearly underway, and from this point work back in time to the point where the signal surpasses the noise level from below for the last time. To this end we define two time-varying thresholds. The upper threshold $u_i$ is used to detect the ongoing transition to the interstadial, and corresponds roughly to a 3-sigma deviation upwards from the mean. The lower threshold $l_i$ is then used to find initiation of

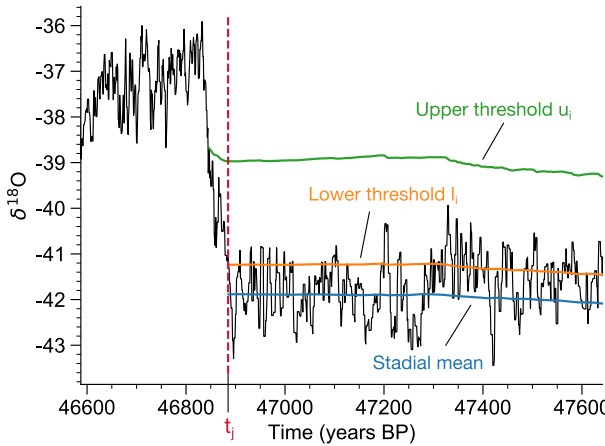

**Figure A1.** Demonstration of the DO onset detection for the case of GI-12 in the stacked record. First, we look for the data point $x_i$ that first exceeds the upper threshold $u_{i-1}$ (green). Next, the data points leading up to $x_i$ are considered, and the point $x_j$ is found, which for the last time crossed the lower threshold $l_{j-1}$ (orange) from below. The onset is then given by the time point $t_j$.

the transition, and corresponds roughly to a 1-sigma deviation upwards from the mean. Both are defined in terms of the mean $\mu_i$ and standard deviation $\sigma_i$ of the entire stadial time series up until data point $i$:

$$u_i = \mu_i + \beta_u \sigma_i$$

$$l_i = \mu_i + \beta_l \sigma_i,$$

where $\beta_{u,l}$ defines the desired deviation from the mean, which is chosen imperially, as detailed below. To define the DO onset, we find the first data point $x_i$ that exceeds the upper threshold $u_{i-1}$. Then, we look back in time and find the last data point $x_j$ that crossed the lower threshold from below, i.e., $x_j > l_{j-1}$ while $x_{j-1} < l_{j-2}$. $x_j$ is then defined as the DO warming onset. This is illustrated in Fig. A1. Note that since the rough location of the DO onset is known, and to avoid false detections in earlier parts of the stadial where mean and standard deviation are still fluctuating significantly, we only start looking for $x_i$

relatively close to the expected onset. As a rule, we start at the stadial data point 80 years prior to the expected onset of the abrupt warming from the piecewise-linear method. For very short stadials, we use a starting point where three quarters of the expected stadial period (determined by the piecewise-linear method) has elapsed.

Since the noise level and the transition amplitude vary between events and ice cores, $\beta_u$ is chosen adaptively. Starting at $\beta_u = 4.5$, we perform the above mentioned routine and check whether the obtained $t_j$ is smaller than a latest reasonable onset time

(the time point of the interstadial maximum). If not, we repeat the procedure with reducing $\beta_u$ by 0.1. $\beta_l$ is chosen empirically according to the noise levels of the records, which result from differences in measurement resolution and accumulation. We choose $\beta_l = 0.5$, $\beta_l = 0.75$, $\beta_l = 0.75$, $\beta_l = 1$ for NGRIP, GRIP, NEEM, and the stack, respectively.

## Appendix B: Sampling-based upper bound on occurrence rate

Here we present a method to obtain an upper estimate of the number of bipolar eruptions in the time period considered here that may be missing in the SVE20 data set, and that would be of similar magnitude. This can be done by using a continuous record of volcanic sulfate deposition from LIN22, which features both the bipolar eruptions of SVE20, and all other volcanic deposition events above a certain deposition threshold found individually in Greenland and Antarctica that could not be identified as bipolar in SVE20. Using this data set assumes that the local deposition in either Greenland or Antarctica is a direct indicator of the global climate impact (magnitude). In reality, there are local eruptions relatively close to the ice core sites, which lead to high deposition in one of the poles, but negligible global climate impact. Thus, any estimate based on these data sets should be considered an upper limit.

Note that we are not aiming to estimate the total frequency of bipolar eruptions of any (arbitrarily small) size, but only those comparable in magnitude to the bipolar eruptions from SVE20, which actually enter our statistical study. These events are above a certain threshold in magnitude, since they need to exceed the noisy background in the glacial ice core impurity records at both poles. Thus, a sensible approach is to estimate the maximum number of eruptions in either of the (unipolar) LIN22 data sets that follow the deposition distribution observed in the bipolar data set (from the same study.) We call the latter the *target* distribution. The sample of unipolar eruptions follows a different distribution, which we will call the *proposal* distribution. The task is to find the largest subsample within the unipolar sample that follows the target distribution. We use a probabilistic procedure, which yields the uncertainty in the size of this largest sample. The unipolar data set is viewed as a particular sample of size $N_u = 501$ from a population with the underlying proposal distribution. We generate further samples of the same size from this distribution (called *proposal samples* hereafter) by drawing randomly with replacement from the unipolar data set. In the following and Fig. B1, we describe how a subsample conforming to the target distribution is found.

We first divide the target sample by quantiles $q_i$, which are defined at (unevenly spaced) percentages $p_i$. Then, we count the number of events $n_i$ in the proposal sample between the target quantiles $q_i$ and $q_{i+1}$, and define the density of events (per percentage point) in this inter-quantile interval by $d_i = n_i(p_{i+1} - pi)^{-1}$. Note that by the definition of quantiles, the target sample of size $N_b$ has the same density of events $d_i = N_b/100$ in all intervals, which is what we are aiming to match by resampling the proposal sample. To do so, we find the interval $j$ with the lowest density $d_j$ and keep all samples in this interval. The remaining intervals contain too many events. To match the density $d_j$, they are resampled by removing each event in the interval with probability $1 - d_j/d_i$. The remaining events form the largest possible subset of the proposal sample that follows the target distribution. Due to the bootstrapping of proposal samples from the original unipolar data set, the number $N_r$ of remaining events is a random variable, and performing the procedure repeatedly with different random samples yields a distribution of $N_r$ from which an average value and confidence interval of the estimate can be derived.

For the data used here, we removed the largest 5 events from the bipolar sample, since these are beyond the range of the unipolar data set. Like this, the target and proposal distributions have roughly the same support. Further, to match the bipolar data set, we remove the eruptions in the interval 16.5-24.5ka from the unipolar data set, as well as those younger than 11.7ka. The following set of percentiles $p_i$ has been chosen: {0, 3, 5, 7, 9, 13, 16, 19, 25, 30, 35, 40, 50, 65, 75, 87.5, 100}. The results

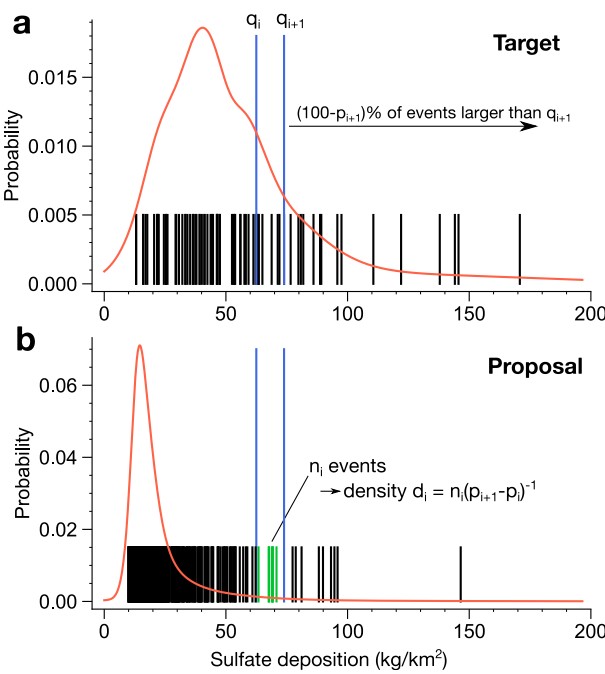

**Figure B1.** Illustration of the resampling method with the Antarctic sulfate deposition data set. **a** Kernel density estimate of the target distribution of the bipolar sample (red), together with the individual events of the sample on the x-axis (black stripes). The quantiles $q_i$ of the sample are calculated for a set of percentages $p_i$. By definition, each inter-quantile interval $q_i$-$q_{i+1}$ contains $N_b(p_{i+1}$-$p_i)/100$ bipolar events, where $N_b$ is the size of the bipolar sample. Note that for a small sample, this holds only approximately. As an example, the 70- and 80-percentiles are shown in blue. **b** The quantiles of the target sample are transferred to the proposal sample (unipolar data set), and the number of events in each inter-quantile interval are counted. Since these are the quantiles of the target sample, the density of events in the intervals is no longer constant. To find the largest subset of events that follow the target distribution, the interval with lowest density $d_j$ is found, and the samples in all other intervals are thinned out to the same density. This is done by removing each event with probability $1 - d_j/d_i$.

are not sensitive to this particular choice. It is only important that, due to the skewed proposal distribution, the quantiles are chosen progressively wider, so that the random proposal samples contain events within all quantiles. The results are given in the main text (Sec. 3.2) and Fig. 4.