# Peer review of "Ice core evidence for major volcanic eruptions at the onset of Dansgaard-Oeschger warming events"

_Climate of the Past, 2022_

## Referee Comment (RC2)

General comments

This paper is proposing and testing a new hypothesis concerning the influence of volcanic eruptions for the onset of so-called Dansgaard-Oeschger (DO) events, which corresponds to large warming events found in Greenland ice cores that takes place in a few decades. For this purpose, the authors first analyse various ice core records from Greenland and Antarctica, which provide estimate of DO variability from δ18O records and volcanic eruptions from concentrations of sulfate deposition in the ice cores. The authors use appropriate statistical approaches to show that volcanic eruptions occur more frequently than random occurrences can explain, less than five decades before the onset of some DO events. This result based on relatively high number of events and well-defined statistical tests seem robust. The authors then analyse the potential impact of such large volcanic eruptions might have on Atlantic Meridional Overturning Circulation (AMOC), a well-known tipping element of the Earth system, which is believed to be associated with DO variability. For doing so, they use an ocean-only model and find that, quite counter-intuitively, a volcanic eruption can reactivate the AMOC from an off-state, through thermally driven large increase of density in the North Atlantic, which allows the AMOC system to cross a threshold and come back to an active state.

This is a very well-presented and structured paper. The science proposed is also well thought and original. The methods are well-depicted and offer first interesting evidences to support the hypothesis presented, as well as interesting physical explanations. For all those reasons, this paper clearly deserves to be published, and fit very well with Climate of the Past main topics. This will be a very valuable input concerning the potential explanations of DO variability.

At this stage, the main weakness of the paper concerns the physical interpretation of the results obtained using ice core reconstructions and statistical analysis. This is because the model used might not be appropriate since it neglects a large number of processes (e.g. sea ice dynamics), include restoring terms at the surface that might strongly affect AMOC dynamics, and use steady state from present-day conditions, which are quite far from the ones during the glacial period. Those shortcomings are relatively well discussed at the end of the paper, but might deserve a bit more explanations for completeness.

Since the authors are honest in their presentation and already highlight relatively well the caveats of their study, I have mainly minor and specific comments to provide to them, which might be useful to further improve the quality of their manuscript.

Specific comments

- Page 6, line 9: I suggest here to also try to discuss the amplitude of the eruptions considered not only as compared to Tambora, but also as compared to Samalas eruption. From what I understand, the mean amplitude of the eruptions considered here to help triggering a DO onset is about 2-3 times larger than that of Tambora, which might correspond about to the amplitude of Samalas (e.g. Jungclaus et al., 2017). There exists a number of models that do consider this eruption in PMIP4 Last

Millennium simulations, which might be interesting to consider in follow up studies to evaluate the associated climate impacts, etc.

- Pages 7, line 19: it should be stated more clearly here that the model is an OGCM. This choice has strong consequences as it is not properly considering interactions with the atmosphere, and more importantly, the restoring terms might strongly affect the stability of the AMOC.
- Page 8, line 9: I assume it is 35 g/kg and not 3.5 g/kg, is that correct?
- Page 9, line 14: the citation of Fig 6a that early in text is raising an issue, since normally, the figures appear in the order of their citation of the text. Also, I have the feeling that too much result materials are presented here, while "Material and Methods" should mainly describe the tools, not the results obtained with them.
- Page 9, lines 8-10: this reference to data from Lin et al. (2021) is a bit surprising at this stage. I think it should be introduced in the section 2.1 and better compared with the other reconstruction used.
- Page 9, line 13: typo: "evidence" should be "evident"
- Figure 3: I suggest to put also estimate from Samalas eruption here.
- Figure 6: as compared to hysteresis from various EMICs shown in Rahmstorf et al. (2005), this one seems a bit different. Present-day state is in particular not bistable, contrary to what is found in a number of AMOC in Rahmstorf et al. (2005). I suggest to discuss this somewhere in the last section and compare the bifurcation figure with Rahmstorf et al. (2005).
- Page 19, line 18: the use of "likely" has a quantitative meaning in climate community due to IPCC reports. I suggest to reformulate this sentence which is not clear enough, notably the use of "some DO warming transitions" into a more quantitative IPCC-like assessment. Given the sentence just before, I would say that: "it is thus *very likely* that volcanic eruptions occurring a few decades before a DO warming contribute to this onset". Indeed, in IPCC terminology "very likely" corresponds to above 90% confidence level. This still does not mean that all DO events are triggered by volcanic eruptions… I let the authors further improve such a formulation to be entirely in line with the high precision of their results.
- Page 20: From my understanding of the impact of large volcanic eruptions on the AMOC, I think numerous processes including sea ice and salinity might be at play. Generally speaking, I would interpret the results obtained from the ice cores a bit differently than what is proposed here using the simple OGCM. Volcanic eruptions are inducing strong excitation of the main variability modes of the North Atlantic (e.g. Swingedouw et al., 2017). In this respect, they can be considered as an exciter, an energy provider of variability of the AMOC following volcanic forcing. Here, following an eruption of the intensity of Samalas, there might large oscillations in the AMOC (e.g. Mignot et al., 2011, their Fig. 2). These oscillations can be positive ones, which clearly might increase the chance of noise-induced bifurcation highlighted here, but interpreted mainly through thermal buoyancy forcing at the surface from the volcanic eruptions, while many other processes might be at play. Thus, it is the fact that volcanic eruptions induce a strong variability (or noise) in the system that explain the shift, whatever the exact processes, which might deserve a more comprehensive climate model. This is more or less already what the authors depict, but this is not stated very clearly in my opinion. The exact mechanisms behind this excitation of larger noise are then not that essential at this stage of the hypothesis (also because

AMOC response to volcanic eruption might be model dependent…even in more comprehensive ones).

References

Jungclaus, J.H., Bard, E., Baroni, M., Braconnot, P., Cao, J., Chini, L.P., Egorova, T., Evans, M., González-Rouco, J.F., Goosse, H., Hurtt, G.C., Joos, F., Kaplan, J.O., Khodri, M., Klein Goldewijk, K., Krivova, N., LeGrande, A.N., Lorenz, S.J., Luterbacher, J., Man, W., Maycock, A.C., Meinshausen, M., Moberg, A., Muscheler, R., Nehrbass-Ahles, C., Otto-Bliesner, B.I., Phipps, S.J., Pongratz, J., Rozanov, E., Schmidt, G.A., Schmidt, H., Schmutz, W., Schurer, A., Shapiro, A.I., Sigl, M., Smerdon, J.E., Solanki, S.K., Timmreck, C., Toohey, M., Usoskin, I.G., Wagner, S., Wu, C.-J., Yeo, K.L., Zanchettin, D., Zhang, Q., Zorita, E., 2017. The PMIP4 contribution to CMIP6 – Part 3: The last millennium, scientific objective, and experimental design for the PMIP4 past1000 simulations. Geosci. Model Dev. 10, 4005–4033. https://doi.org/10.5194/gmd-10-4005-2017

Mignot, J., Khodri, M., Frankignoul, C., Servonnat, J., 2011. Volcanic impact on the Atlantic Ocean over the last millennium. Clim. Past 7, 1439–1455. https://doi.org/10.5194/cp-7-1439-2011

Rahmstorf, S., Crucifix, M., Ganopolski, A., Goosse, H., Kamenkovich, I., Knutti, R., Lohmann, G., Marsh, R., Mysak, L.A., Wang, Z., Weaver, A.J., 2005. Thermohaline circulation hysteresis: A model intercomparison. Geophys. Res. Lett. 32, L23605. https://doi.org/10.1029/2005GL023655

Swingedouw, D., Mignot, J., Ortega, P., Khodri, M., Menegoz, M., Cassou, C., Hanquiez, V., 2017. Impact of explosive volcanic eruptions on the main climate variability modes. Glob. Planet. Change 150, 24–45. https://doi.org/10.1016/j.gloplacha.2017.01.006

---

## Author Response (AR1)

**Referee #1**

Below are our responses to the issues raised by the referee (*italic*).

*A possible issue might be that it seems to me (apologies if I have missed this – if I have, it needs to be made much clearer) that all identified bipolar eruptions were treated equally rather than separated out by hemisphere. Many studies now show that the latitude of an eruption greatly affects the nature of the response, so this really needs to be considered. Lin et al 2021 has published the estimated latitudinal band of eruptions across the interval 60-9 ka, so this should be easily done. I think that this is particularly problematic in terms of the conclusion that stadial events are not triggered by volcanism; this really should look only at NH eruptions. Also, it seemed to me that all stadial onsets were used in the statistics, regardless of the type, rather than just abrupt stadial onsets. These points are elaborated on in the comments below.*

We thank the referee for these points, and our detailed answers follow the more specific points by the referee below. In short, the latitudinal estimates of Lin et al are not precise enough to base our objective analysis on them. We do not believe it would be a wise choice of methodology to restrict our analysis to a very uncertain subset of eruptions based on a preconceived idea of a physical mechanism for the potential volcanic trigger of DO events, as the referee suggests.
Regarding the stadial onsets, it does not change the results whether only truly abrupt events are considered. There are simply not enough eruptions occurring shortly before these stadial onsets to give significant results, regardless how small a subset of stadial onsets one chooses.
Nevertheless, these points will be discussed in the revised manuscript, and explained in detail below.

*The manuscript states that there was one bipolar eruption about every 500 years. A key paper that needs to be discussed is Rougier et al., 2018 EPSL, where they calculate probable return periods for eruptions of different magnitudes. Rougier et al estimate that there was a M6 eruption every 110 years, and an M7 eruption every 1,200 years; both magnitudes would be sufficient to have a bipolar expression (e.g., the M6 1991 Pinatubo eruption was not particularly large, but did result in SH S deposition, Cole-Dai et al., 1999 and others). This should be discussed in the manuscript, as the return period for M6 bipolar eruptions is far shorter than 500 years. The authors do appeal to ice thinning and a higher background of impurities, but even if the eruptions cannot be detected for these reasons, they still presumably happened, and not all of these will be M6 eruptions; some high latitude M7 eruptions may be missing from the opposite hemisphere if the authors' contention of ice thinning is correct.*

We are happy to mention and discuss the return times for M6 and M7 eruptions obtained in Rougier et al, and will do so in the revised manuscript. Note however that these return times are based on geological evidence of the erupted mass, which is not directly calibrated to the sulfur deposition estimate that is used in our work. To make direct comparisons, one has to rely on individual known eruptions that are in both data sets. This would again be eruptions like Tambora or Samalas that we already compare our data set to (the latter in the revised manuscript), albeit in the context of Sigl et al 2015.

Our data is largely consistent with the return period of these eruptions, which however have large uncertainties both in Sigl et al and Rougier et al, due to the very short time periods investigated. Note here that while Rougier et al do consider data spanning 100 kyr, their estimates for the occurrence rates of eruptions with magnitude M<7.5 are based only on the last 2 kyr.

Referring again to Sigl et al 2015, we already mention in the manuscript that there are indeed many more eruptions with a bipolar signature that are however smaller than what can be detected in the glacial period. This leaves us (Svensson et al 2020) naturally with a data set containing bipolar eruptions of larger magnitude. The smaller, undetected eruptions with a bipolar signature are not relevant to our analysis, since a) they are less likely to have global climatic impact, and b) they would need to be assumed a priori to occur both in the vicinity of DO onsets and elsewhere in time. In other words, our statistical framework does not depend on the knowledge of these events.

The fact that some larger eruptions (e.g. high latitude M7 eruptions) could be missing in our data set is covered and quantified extensively in the manuscript, where we test the robustness of the results against a scenario where more than 50% of the relevant eruptions would be missing.

Finally, note that we strongly prefer to compare our data to Sigl et al, since the Rougier data is incomplete in that it only covers eruptions of known source. From the data in Sigl et al we can see that 12 out of the 25 largest eruptions (as detected in ice cores) of the last 2,500 years are in fact of unknown source.

*The article overall is written somewhat awkwardly and is difficult to follow in places. Overall, the text could be simplified and shortened considerably.*

We are happy to hear specific suggestions. For the time being we will try to improve the clarity of the text and try to make it more concise.

*P1, L1-2: Vague as written – across what timescales? It is fairly well understood now over the past 2,000 years or so, so please be clear what timescale you are referring to.*

We are referring to pretty much all timescales actually, since volcanism has been suggested as a driver of the climate in many different climatic periods. We will write this more precisely, also in relation to the following point.

*P1, L2: The statement regarding a statistical assessment being hampered is incorrect – Baldini et al and Bay et al are both statistical assessments. Perhaps rephrase to include reference to make reference to the particular issue here: that it is difficult to work out the magnitude from S concentrations in one ice core alone, and difficult to correlate individual spikes across Greenland and Antarctica. I think that the 'statistical assessment' is meant to refer to this.*

We will write more specifically what we think is required for a "statistical assessment", as explained in the following: We do not mean that statistical assessments have not been conducted, but that they were challenged by poor data quality/very large temporal uncertainties (several centuries) that made it difficult to actually argue for a causal relationship of eruptions and climate. In previous work one simply did not know whether an eruption occurred before or after an abrupt climate transition. Further,

as the reviewer correctly states, relying on deposition estimates from one of the poles potentially introduces many local eruptions of limited global impact into the data set.

*Page 1, 17: 'Greenhouse' does not need to be capitalised.*

Ok.

*Page 3, line 18: I believe that this submission is a heavily modified version of a previous unsuccessful submission to Climate of the Past. I note that one of the improvements is the inclusion of the Lin et al 2021 dataset of eruption magnitude.*

Indeed.

*Page 3, line 18: 'volcanogenic sulphur deposition' rather than 'volcanic depositions'*

Ok.

*Page 3, line 8-28: this seems like too much summary and interpretation for the introduction. A short 'here we address this' or 'here we do this to show that' is fine, but two long paragraphs with interpretation is too many for the end of the intro.*

Ok, will rewrite.

*Page 5, L25-30: Can a comparison with the absolute Corrick et al (2020) dates help say something about accuracy? Noting of course that here the interpretations are based on proxies from the same cores (so absolute timing is less important).*

As the reviewer states, the absolute ages are not relevant to the analysis, and systematic offsets with respect to absolutely dated archives are not the subject of this study. Since these offsets are much larger than a) the relative age uncertainty of single onsets across cores, and b) the time windows analyzed for the co-occurrence of volcanic eruptions, we do not think that a reference to Corrick et al (2020) is of any help. The precision of our onset estimates cannot be judged because the uncertainties in the Corrick et al ages are 1-3 centuries. Further, we are estimating the onsets of the transitions, and not the midpoints as in Corrick et al.

*Page 5, L29-30: unclear what is meant here. Perhaps rephrase to (what I think is the intended meaning): 'It is therefore possible to assess the climate repercussions of volcanic eruptions to a decadal-scale.'*

Indeed, this is what we mean. Will rephrase.

*Section 2.3: p6, L1-18: This discussion is welcome, and represents a key change from the previous submission. I think that it should be made clear though that many of the other sulphate peaks are probably not just noise, but recording smaller regional eruptions, or even bipolar eruptions that are not conclusively matched to the other hemisphere.*

Sure, we will write this explicitly.

*P7, L1-3: This conflicts with the findings of Zielinski et al., 1997 JGR, who found that eruption frequency increased during deglaciations, possibly due to crustal stresses. This should be mentioned and reasons for any differences discussed.*

We agree that this is worth discussing in some detail. Still, our statements do not contradicts the findings. In the revised manuscript, we will clarify the following points, adding to Section 2.3. First, Zielinski et al do not actually show that the increased volcanic activity during the deglaciation is significant, and this would also be difficult for them given the poor data quality at the time. Second, we are specifically mentioning and showing that the frequency is indeed elevated. A more detailed analysis can also be found in Lin et al. 2022 (Fig. 3+4), where it is seen that the increase in eruptive activity seems to mostly concern increases in the magnitude of large eruptions as detected in Greenland.

Nevertheless, restricting ourselves to the bipolar dataset from Svensson et al 2020, our analysis shows that in the bigger scheme of the glacial period the deglacial increase of eruptions does not appear as statistically significant, even though it may well be a genuine feature. Given that it is not significant there is no sound basis for us to include this in our null model. Apart from this, since a) it is only a very short segment of the whole time period, and b) we already test our results against severe undercounting of events, this does not influence our results.

*Section 2.4: The model simulation is okay to include, and there appear to be adequate caveats in this section regarding how comprehensive it is. It is still meaningful to show that volcanism can trigger oceanic changes in at least one model. But given the limitations of the model (see below) more caveats in the abstract/conclusions are probably warranted.*

Ok, will try to make it more clear in the abstract and conclusions.

*P7, L27: How realistic is the use of present-day ERA-40 wind stress forcing? I believe that the re-analysis data extends from 1957 through August 2002, during the anthropogenic greenhouse era; surely the winds during stadials would be considerably different?*

Indeed the boundary conditions are present-day and not realistic for the last glacial. We emphasized the wind stress here and wrote "realistic" because it was one of the changes to the model configuration in Lohmann and Ditlevsen 2021. We will drop the word "realistic".
Note that within this class of models no "realistic" stadial boundary conditions exist unfortunately, since the stadial state of the atmosphere is not well-constrained and very few coupled ocean-atmosphere models can simulate a stadial climate. As a result, the stability of the AMOC is very much unconstrained during the last glacial (and in fact also in present-day).

In our hypothesis we assume that an instability of the AMOC (and corresponding stable states analogous to stadials and interstadials) exists, and achieve this instability by changing the freshwater boundary conditions. We are aware that changes in the wind stress would again change the AMOC stability. We will write this more explicitly in the revised manuscript (Sec. 2.4).

*P7, L33: That the model does not including a sea ice or atmospheric component could be a major issue, because there is a good possibility that sea ice plays a major role in any positive feedback mechanism. This might not be a 'fast' amplifying feedback, but could potentially affect climate through the duration of the event. Again, I have no major issue with the model being included, but it needs to be adequately caveated.*

We agree that sea ice and atmospheric components have the potential to change the mechanism. But we find it hard to speculate how e.g. sea ice will affect the dynamics without actually modeling it. This will for example depend on how large the actual sea ice extent is during the different stadials. We will write in the revised manuscript (Sec. 2.4 and Discussion) that our proposed mechanism is contingent on the sea ice and atmospheric response to not destroy the dense water formation in the North Atlantic.

*P8, L30: again, when saying "We have estimated the precise times of the DO warming onsets….." it is well worth comparing with the Corrick et al onsets, to briefly compare the accuracy of the derivations here.*

See response further above. The relative timing of events is precise in our study because we are working on records from well synchronized ice cores. But the absolute ages are inaccurate because the dating is based on annual layer counting. Because it is the relative timing of volcanic eruptions and climate transitions that matters for our study, high accuracy is however not crucial.

*Page 9, L16: define which 'this study' means. Is it Lin et al., as mentioned in the last sentence, or this current study?*

We mean Lin et al, will state explicitly.

*Page 10, L1-3: Why choose the 5 largest of the last 2,500 years? If the 10 largest were chosen, then the observed frequency would be 250 years instead of 500. What is the justification for choosing the largest 5?*

We choose the 5 largest events because 2500/5 = 500 years, and 500 years is the return period of the events in our data set. The magnitude of the 5 largest events of the last 2,500 years matches the magnitudes of the bipolar glacial data set in the sense that most eruptions in bipolar data set fall in the same magnitude category (larger than Tambora). Note that this is only a qualitative comparison to illustrate that the frequency and the estimate magnitudes of the bipolar dataset are consistent with what we would expect based on datasets of more well-known and well-studied eruptions. Talking for instance about the 10 largest events with a return period of 250 years would be less relevant, since we specifically want to make a statement about 1-in-500 year eruptions.

*Section 3.2: This section really needs to reference and utilise the information available in Rougier et al., 2018, EPSL, where they provide estimates of the recurrence of eruptions of different magnitudes.*

See response further above. Again, the Rougier et al. study is based on geological evidence of known eruptions that is likely to miss out events at an increasing rate the further back in time one goes.
In fact, even in the period of the last 2.5 kyr, the data (of the largest eruptions) analyzed by Rougier et al is likely sparse. In the 2015 ice core study by Sigl et al, it is found that half of the largest 25 eruptions are of unknown source, and thus unlikely to be found in the data sets of Rougier et al. Accordingly, the estimated return periods of prominent eruptions such as Tambora are significantly larger in Rougier et al compared to what can be inferred from Sigl et al. Thus, we believe the ice core record allows for a more consistent detection of large volcanic eruptions over the investigated period.

*Section 3.6.: I agree that the drops back into stadial events are much less well defined as the onset of rapid warming. However, I disagree with how these are handled in this manuscript. Lohmann and Ditlevsen (2019) identified the ends of interstadials, and it is these data that are used here. However, (as far as I understand) in this present manuscript all these interstadial dates are used, regardless of whether or not it was a sudden transition. It is clear that the trajectory and duration of the warm phase of many DO events is predictable based on linear extrapolation, and therefore the ends of these particular events should not be considered as 'events' in the calculations. Rather, only the sudden drops in some events (such as in DO-20 and 19.2) that deviate from the predicted linear trend could have been caused by volcanism. This could affect the statistics. I would suggest either only including DO events ending with sudden drops, or not considering the ends of DO events at all, and only focussing on their initiation.*

We understand the referee's concern that the missed statistical link of terminations and eruptions may be due to the too large number of terminations that are not abrupt and thus not good candidates for a volcanic trigger.
However, first of all, we argue that in principle all events are candidates for a volcanic trigger, regardless of whether they can be predicted approximately from linear trends. In Lohmann/Ditlevsen 2019 Clim Past, we show that the linear oxygen isotope trends at the beginning of the interstadials can be considered a good predictor of the eventual stadial onset. In Lohmann 2019 (GRL) the same is shown more rigorously for the stadials (i.e. interstadial onsets), using dust records. But the predictions of the events are only approximate and thus leave room for a short-term volcanic trigger, regardless of whether the rough timing of the event may be set from the beginning by some other processes. This is discussed in detail in the manuscript, and it is the basis of our interpretation of the results, i.e., why there can be a volcanic trigger of (only) some events even though they are predictable in principle.

Regarding the stadial onsets, we do agree that it is a priori more obvious to look for a potential trigger in the "well-defined" or abrupt onsets. And indeed one might argue that there are quite a few stadial onsets that are simply not well-defined enough to even look for a potential trigger, at least when considering the oxygen isotopes alone (as in Lohmann/Ditlevsen 2019). It is not trivial how to best define objectively which events are abrupt and which are not. A procedure is proposed in Lohmann/Ditlevsen 2019, which yields that 10 out of the 19 terminations regarded in the present paper are not abrupt.

Regardless of whether the referee agrees with all that was said previously, the fact that we do not see a significant relationship of terminations and eruptions is not because we are using too many terminations that are no good candidates for a volcanic trigger:

As can be seen in Fig. S5, there is really only one interstadial where a volcanic eruption happens within 100 years prior to the interstadial termination.
So this is clearly not significant, even if the number of admissible events is reduced to half or even less. In fact, if we were to only use the events that were identified as abrupt in Lohmann/Ditlevsen 2019, there would be no termination where the closest preceding eruption occurs with 170 years prior. Thus, we will not be able to find any subset of DO events where a significant link of terminations and eruptions exists, no matter how much cherry-picking we allow.

We do not claim that this is a proof for the absence of such a link. Better data may show otherwise. Besides unidentified bipolar eruptions, the most obvious candidate for a "missed" statistical link of interstadial terminations and eruptions is the large uncertainty in the timing of the terminations, as discussed in the main text. Specifically, some of the interstadial terminations may be identified too early with our method. We'll discuss all of this explicitly in the revised manuscript (Sec. 3.6).

*P19, L19: The sentence here makes it seem that the authors of this manuscripts are the first to detect a volcanic influence on DO event onset, whereas both Baldini et al and Bay et al also did. I would rephrase. One suggestion is:*

*"Thus, we conclude that there is a likely influence of large volcanic eruptions on the occurrence of some DO warming transitions, consistent with the results of previous studies (Bay et al., 2004, 2006; Baldini et al., 2015), but do not find evidence for a similar statistical relationship of eruptions preceding the abrupt DO cooling transitions." However – note that the statistics concerning the onset of the cooling phase could be incorrect as outlined above, and therefore the last part of the sentence above may need to be deleted in a revised submission.*

We will rephrase to make it less ambiguous. There are both agreements and disagreements to the two previous studies, which we will state more explicitly. Regarding the stadial onsets, see the previous point.

*Additionally, both Baldini et al and Bay et al looked at hemisphere-specific eruptions, whereas this manuscript does not (apologies if I have missed this – if it is there it needs to be much more clearly stated). Many recent papers covering the more recent past note a different response between NH and SH eruptions, so that this really needs to be considered. For example, Zhuo et al 2021 (Atmos. Chem. Phys.) use two groups of ensemble simulations to show how NH, equatorial, and SH eruptions trigger very different climate responses, including ITCZ migration away from the hemisphere of the eruption in the case of high latitude eruptions. Sun et al 2019 (J. of Clim) argue that NH high latitude eruptions could affect ENSO state, providing another example of why latitude and hemisphere are important.*

*Specifically, it may be that NH eruptions trigger abrupt stadial onsets, and that because the statistics here consider all eruptions (NH or SH), this link was missed. There are few lines from 23-29 that mention hemispheric asymmetry, but it needs to be clearer if the statistics do take this into account, and, if not, then either it does need to be considered or the section about stadials removed. Lin et al 2021 do divide out the eruptions according to estimated latitudinal band, so perhaps this information could be used.*

We agree that latitude is important for the climate response and will include the Zhuo et al 2021 and Sun et al 2019 references in the revised paper to further emphasize this point (Sec. 3.7 and Discussion), expanding what is already discussed and investigated with our model simulations.

Indeed, Lin et al give a binary classification (below 40degN or Southern Hemisphere, LLSH; versus Northern Hemisphere High Latitude, NHHL) of the eruption latitude based on the relative Greenland-Antarctic deposition. We do not believe, however, that this data is informative and reliable enough to incorporate it into our statistical analysis, or even use it as the main data of the analysis.
We will however discuss it in the revised manuscript (Sec. 3.7 and Discussion), in relation to our proposed mechanism of the volcanic trigger, giving the following details:

There are 5 (LLSH) vs 2 (NHHL) eruptions within 50 years before a DO onset, and 4 (LLSH) vs 1 (NHHL) eruptions within 20 years. This is out of 34 (LLSH) vs 48 (NHHL) total bipolar eruptions.

While the sample size of eruptions before DO onsets is very small, it still indicates that eruptions classified as LLSH are more likely to cause a DO warming. We understand this could be taken as evidence that our hypothesis of a North Atlantic (NA) cooling to trigger the DO onsets is not so likely.

However, the climatic impact and its latitudinal footprint is still very much unconstrained, since the classification of Lin et al 2022 only tells us whether there has been relatively more deposition of sulfur in Greenland versus Antarctica.

The first obvious impediment is the large uncertainty in the deposition estimates of individual eruptions, due to inter-core variability and other factors. But even if this uncertainty was zero, there is still a massive uncertainty in the climatic impact, because we do not know where most of the sulfur aerosols actually were transported (troposphere or stratosphere) and how long they were present in the atmosphere. So there is the possibility that an eruption with a large sulfur deposition in Greenland compared to Antarctica still produced less NA cooling than a tropical eruption classified as LLSH.

Further, the latitudinal estimate "LLSH" includes NH eruptions up to 40N. So based on this classification we do not have a clear separation of eruptions with and without a NA climate impact. Note that even tropical eruptions tend to have a hemispherically asymmetric cooling towards the NH, as discussed in the manuscript.

All in all, the latitude classification by Lin et al 2022 does not warrant us to focus the analysis on either latitudinal subset. Even more so because there is no other evidence for our NA cooling hypothesis that would warrant us to have the hypothesis inform the data analysis.

We will, however, include a detailed discussion on this in the manuscript (adding to Sec. 3.7 and the Discussion), so the reader can decide to what degree this is in conflict with our physical hypothesis.

**Referee #2**

Below are our responses to the issues raised by the referee (*italic*).

*Page 6, line 9: I suggest here to also try to discuss the amplitude of the eruptions considered not only as compared to Tambora, but also as compared to Samalas eruption. From what I understand, the mean amplitude of the eruptions considered here to help triggering a DO onset is about 2-3 times larger than that of Tambora, which might correspond about to the amplitude of Samalas (e.g. Jungclaus et al., 2017). There exists a number of models that do consider this eruption in PMIP4 Last Millennium simulations, which might be interesting to consider in follow up studies to evaluate the associated climate impacts, etc.*

Thank you for this suggestion. We specifically chose a comparison with 1-in-500 year eruptions (greater or equal Tambora), because the return period of eruptions in our data set is actually 500 years. The return period of a "Samalas-magnitude" eruption in our data set is also consistent with the expected return period of a Samalas-type eruption estimated previously from ice cores, which is however less certain then a Tambora-type. In the revised manuscript (Sec. 2.3 as well as Fig. 3) we will include a comparison with the Samalas eruption.

*Pages 7, line 19: it should be stated more clearly here that the model is an OGCM. This choice has strong consequences as it is not properly considering interactions with the atmosphere, and more importantly, the restoring terms might strongly affect the stability of the AMOC.*

We will be more specific in saying that it is ocean-only, and that processes not considered in the model may alter the stability of the AMOC. Note, however, that the stability of the AMOC both in present-day and the glacial is still very much a matter of debate and very model-dependent. Thus, the point here was to propose a mechanism where we simply assume bi-stability of the AMOC, and to use an ocean model that fulfills this assumption. We will state explicitly in the revised manuscript that our proposed mechanism is contingent on the sea ice and atmospheric response to not destroy the bi-stability of the AMOC and the dense water formation in the North Atlantic as a volcanic trigger of an AMOC transition.

*Page 8, line 9: I assume it is 35 g/kg and not 3.5 g/kg, is that correct?*

Indeed, thank you.

*Page 9, line 14: the citation of Fig 6a that early in text is raising an issue, since normally, the figures appear in the order of their citation of the text. Also, I have the feeling that too much result materials are presented here, while "Material and Methods" should mainly describe the tools, not the results obtained with them.*

Ok. We will move some material from Sec. 2.4 to the main text.

*Page 9, lines 8-10: this reference to data from Lin et al. (2021) is a bit surprising at this stage. I think it should be introduced in the section 2.1 and better compared with the other reconstruction used.*

We assume that with "other reconstruction" you mean the Svensson et al 2020 data set? We will add a Section 2.2. where we better describe the two volcanic data sets. Section 2.1 is dedicated to the isotopic data sets.

*Page 9, line 13: typo: "evidence" should be "evident"*

Ok.

*Figure 3: I suggest to put also estimate from Samalas eruption here.*

Ok.

*Figure 6: as compared to hysteresis from various EMICs shown in Rahmstorf et al. (2005), this one seems a bit different. Present-day state is in particular not bistable, contrary to what is found in a number of AMOC in Rahmstorf et al. (2005). I suggest to discuss this somewhere in the last section and compare the bifurcation figure with Rahmstorf et al. (2005).*

We are happy to briefly discuss this in the revised manuscript (Sec. 3.7). A detailed study of the bifurcation structure and the landscape of stable and unstable states in the model is currently underway. The main feature of the model that is important here is that there is a tipping point of the AMOC. The shape and width of the hysteresis loop depends on boundary conditions and other very much unconstrained parameters in the context of ocean-only models. We don't think it is at present possible to argue which ocean-only model or EMIC gives an accurate representation of the AMOC stability in the glacial period.

*Page 19, line 18: the use of "likely" has a quantitative meaning in climate community due to IPCC reports. I suggest to reformulate this sentence which is not clear enough, notably the use of "some DO warming transitions" into a more quantitative IPCC-like assessment. Given the sentence just before, I would say that: "it is thus very likely that volcanic eruptions occurring a few decades before a DO warming contribute to this onset". Indeed, in IPCC terminology "very likely" corresponds to above 90% confidence level. This still does not mean that all DO events are triggered by volcanic eruptions… I let the authors further improve such a formulation to be entirely in line with the high precision of their results.*

Thank you for this suggestion, we agree that the sentence is vague and will adapt accordingly.

*Page 20: From my understanding of the impact of large volcanic eruptions on the AMOC, I think numerous processes including sea ice and salinity might be at play. Generally speaking, I would interpret the results obtained from the ice cores a bit differently than what is proposed here using the simple OGCM. Volcanic eruptions are inducing strong excitation of the main variability modes of the North Atlantic (e.g. Swingedouw et al., 2017). In this respect, they can be considered as an exciter, an energy provider of variability of the AMOC following volcanic forcing. Here, following an eruption of the intensity of Samalas, there might large oscillations in the AMOC (e.g. Mignot et al., 2011, their Fig. 2). These oscillations can be positive ones, which clearly might increase the chance of noise-induced bifurcation highlighted here, but interpretated mainly through thermal buoyancy forcing at the surface from the volcanic eruptions, while many other processes might be at play. Thus, it is the fact that volcanic eruptions induce a strong variability (or noise) in the system that explain the shift, whatever the exact processes, which might deserve a more comprehensive climate model. This is more or less already what the authors depict, but this is not stated very clearly in my opinion. The exact mechanisms behind this excitation of larger noise are then not that essential at this stage of the*

*hypothesis (also because AMOC response to volcanic eruption might be model dependent…even in more comprehensive ones).*

We agree in principle with this interpretation, and with the fact that the exact processes that occur after the volcanic eruption and an excitation of the AMOC deserve a more comprehensive model. We will stress this better in the revised manuscript.

The point of our specific proposed mechanism, as opposed to a more general excitation of AMOC variability, is that it may explain why the eruptions seem to excite AMOC resurgences and not shutdowns. We are of course aware that our explanation is only one of many others that would also involve other processes in sea ice and atmosphere. We will state more clearly that the eruptions may also just lead to a general excitation of AMOC variability, and that this could lead to a spontaneous AMOC resurgence. We will also more clearly point out the varied impacts of different types of eruptions (latitude, season etc.) on the AMOC, referring to Mignot et al 2011.

**Referee #3**

Below are our responses to the issues raised by the referee (*italic*).

*The validity of the statistical methods relies on two key points:*

- *The bipolar volcanic catalogue by Svensson et al. (2020) is complete, and*
- *The choice of Null hypothesis (1 eruption per 500 years) is correct*

*I am not convinced that the authors have demonstrated either.*

We agree with the referee that in order for our conclusions to be reliable, the true rate of eruptions corresponding to the magnitude of the eruptions observed shortly before DO onsets needs to be known. The "two key points" identified by the referee are actually two ways of stating the same thing, since an "incomplete" volcanic catalogue would simply translate into an underestimate of the eruption rate.

As discussed in more detail further below, in the paper we give a number of arguments (based on other data sets) supporting that our null hypothesis is suitable and the statistical analysis can be applied to the data set at hand. We indeed allow for the possibility that the volcanic catalogue is incomplete and that the true eruption rate is higher.

We give quantitative constraints on how far the data might in fact be biased in this way, and show that the results are still significant when taking this into account. In addition, in the revised version of the paper (and in this author response) we give further arguments to support this, and hope that this can convince the referee of the validity of our method and conclusions.

*(1) the authors rely on the bipolar events identified by Svensson et al. (2020) using annual layer counting. The goal of that study was not to make an unbiased and complete catalogue of all bipolar events, but rather to to investigate the phasing of bipolar climate change at great precision. Svensson*

*et al (2020) write: "The bipolar layer counting is not continuous but is focused on periods of abrupt climate variability or high volcanic activity."*

*This implies that the Svensson et al. (2020) catalogue is biased toward periods of abrupt climate change. This is critically important. LS22 demonstrate convincingly that there are more events in the vicinity of the abrupt events, however does this reflect the DO triggering mechanism proposed, or simply a bias in the volcanic catalogue towards periods of abrupt climate change that were investigated in more detail (as suggested by Svensson 2020)?*

*It seems critical to me that the bipolar layer counting has to be done continuously across the full interval of interest with no regard to the presence of abrupt transitions, and not just the periods of abrupt climate change.*

*Anders Svensson is an author on both studies, and I am interested to hear his perspective on this issue.*

Hello, this is Anders Svensson speaking.
The main reason for stating that the layer counting of Svensson et al, CP, 2020 is not continuous is to say that there is no new time scale provided with that study. Had the layer counting been covering the 12-60 ka interval continuously it would at the same time have been a revision of the GICC05 time scale.
In the right hand side of Table S2 of Svensson et al., CP, 2020, the layer counting made for that study is shown in column 'This study.' As mentioned in the paper, the section 16.5-24.5 ka b2k is not covered by the study as ice cores are difficult to synchronize in this interval. Apart from this gap most of the 12-60 ka period is, however, counted continuously in Svensson 2020. There are four gaps in the layer counting at 14.5, 27.8, 32.0 and 37.1 ka, respectively (indicated as blank fields in Table S2), spanning a total of some 6500 years or about 16.5% of the investigated 12-60 ka period (excluding the 16.5-24.5 ka section.) Thus for more than 80% of the investigated period layer counting has been performed and the major bipolar volcanic eruptions occurring in those intervals have been identified.

In Svensson 2020 Figure 1 the identified bipolar eruptions are indicated. Except for the periods where no layer counting has been performed, the eruptions are fairly evenly distributed in time. (Note that the high density of bipolar events in the GI-9-10-11 interval is due to the five Laschamp-related 10Be bipolar matches that are also included in the figure.)

Nevertheless, the statement from Svensson 2020 only says that the layer counting was not done continuously. While the layer counting was indeed done in segments that comprised clusters of eruptions that could be matched as bipolar, it does not imply that the actual procedure of finding bipolar matches was not done on the whole time period as continuously as possible. Since the statement goes on to say that it focused on "periods of high volcanic activity", it should be clear that periods with large eruptions were considered, regardless of whether they happened close to DO events.

That being said, while an effort was made in Svensson 2020 to provide a bipolar volcanic data set that is as continuous as possible, it is indeed quite likely that not all of the large bipolar eruptions have been identified, especially in the intervals of no layer counting, but potentially also in some of the longer layer counted sections, where the synchronization may have failed.

This is the reason why we added a comprehensive analysis of the continuous data set of Lin et al since our first submission to Clim Past. This gave upper bound estimates on the number of potentially

missing bipolar eruptions. Based on these estimates we allowed for up to two times more large bipolar volcanic eruptions in our robustness tests of LS22 as compared to those identified in Svensson 2020.

Here we want to stress that we have several lines of evidence to support that the dataset is in fact relatively complete, and certainly suitable for our analysis:

1. The magnitude of (most of) the eruptions, including the ones before DO events, is consistent with eruptions larger than Tambora, and thus with the observed 500 year return period. Unless the frequency of eruptions of that size was significantly larger in the glacial, there is thus no reason to suspect we are missing a large number of eruptions.

2. Even without reference to known eruptions with relatively well-known return periods like Tambora, we can constrain the number of eruptions that may be missing from our dataset, using a different, fully continuous and objective list of Greenland and Antarctic volcanic eruptions provided by Lin et al 2022. This yields our upper bound that the true occurrence rate may be up to 1.5-2 times higher. It is an upper bound, since it also includes a large fraction of regional eruptions with large deposition on only one of the poles. The results are still significant when using this rate.

3. The magnitude of the eruptions before DO events is a representative sample of the magnitudes of all of the bipolar eruptions. Thus, there is no evidence that relatively smaller eruptions (with corresponding higher recurrence rate) have been picked specifically in the vicinity of DO onsets in order to get the sought-after bipolar phasing of DO events.

4. This point is new and will be included in the revised manuscript (Sec. 3.1 and new panel in Fig. 2): In general, there is no clustering of eruptions around DO events, since there are very few cases where an eruption happens shortly after a DO onset. With "very few cases" we mean here "not more than would be expected by chance", i.e., in our data we find 0 eruptions within 20 years and 2 eruptions within 50 years after the onset. This is opposed to 5 eruptions within 20 years and 7 eruptions within 50 years before the onset.
From the point of trying to find a bipolar match to constrain the bipolar phasing of DO events (as a potential source of bias in our dataset), a match shortly after a DO onset would be just as useful as a match shortly before. Further, a match shortly after the interstadial onset should even be easier to establish due to the higher accumulation rate and lower impurity noise.

5. Apart from the occurrence of 5-7 eruptions closer to DO onsets than would happen by chance, there is no evidence of general clustering around the times of abrupt change, since the waiting times in between events are consistent with a stationary Poisson process. This would not be the case otherwise, as it would create gaps that lead to a tail in the distribution that is not consistent with the exponential distribution.

6. Note that there are more than 70 eruptions not close to DO events. First, this should illustrate that it was not the sole purpose of Svensson 2020 to find eruptions close to DO events. Second, these eruptions were identified in patterns of typically ~4 eruptions, which are spaced by 500 years on average. Thus, even if Svensson 2020 would have only looked at the DO onsets, the resulting volcanic data set would already cover 20*3*500 = 30,000 years (or more depending on how one interprets the edges of the eruption patterns) out of the 40,000 total years. Thus, by definition of the matching via patterns of eruptions and the given spacing of eruptions and DO events, this necessarily yields a volcanic record with fairly complete coverage even if only the vicinity of DO onsets were allowed as starting points (which was not the case).

Regardless of all this, we do want to reiterate that the dataset does not even need to be fully complete/continuous, as long as we can constrain the true rate of eruptions, and show the eruptions before DO events in question are of the correct magnitude for the given rate, both of which we do in the manuscript. We do admit that the manuscript in the present form did not name several of the potential biases and concerns explicitly, which will be added in the revised manuscript (Sec.'s 2.3 and 3.2, as well as Discussion) along with the more detailed reasoning above.

*(2) the choice of the null hypothesis is critical here. The authors base their choice on the assumption that the Svensson 2020 bipolar catalogue is complete, which yields 1 large bipolar event per 500 years.*

*(2a) However, the quote from Svensson (2020) implies that the catalogue is not complete, but instead focused on periods of abrupt climate change or large volcanic acitivty. The recurrence time of such events should thus be shorter.*

See previous comment.

*(2b) It is clear that the Svensson catalogue has more events during the deglaciation (12-16 ka) than during the glacial (24-60 ka). Performing a simple Student t-test on the recurrence time distribution of these two intervals will show this I believe. This makes intuitive sense given the larger annual layer thickness in the former interval. What would happen if the deglacial recurrence time was used as the basis of the Null hypothesis? There are also many bipolar links identified in the Holocene (e.g., AICC2012). What is the recurrence time of those?*

We already perform statistical tests on whether the deglacial eruption frequency is higher and report this in the manuscript. Taken individually, indeed two 1,500 year periods have a significantly higher occurrence rate than the whole population (at 90% confidence). However, in the larger context of the glacial, this is not statistically significant due to multiple testing.

Our analysis thus shows that in the bigger picture of the glacial eruptions this increase of eruptions does not appear as statistically significant, even though it may well be a genuine feature. Given that it is not significant there is no sound basis for us to include this in our null model.
This is even more so since the increased frequency may well be an artifact of the higher measurement resolution of the proxies during the deglaciation (as you say), thus allowing one to observe eruptions of smaller magnitude (and thus higher frequency), compared to the rest of the glacial. On top of this, using the observed frequency during the deglaciation would not be a good null hypothesis since a) it is a very small sample to reliably estimate a recurrence time, and b) there are enough potential physical explanations for why the frequency of eruptions during the deglaciation may not be representative of the whole glacial.

Further, since a) the deglaciation is only a short segment of the entire time period, and b) we already test our results against severe undercounting of events, this potential issue does not influence our results. We will still include a better discussion of it in Sec. 2.3 of the revised manuscript.

Regarding the recurrence time of eruptions in the Holocene, this is already mentioned in the paper on page 6. There we write that 80 eruptions with bipolar signature have been found in the last 2,500 years

in Sigl et al 2015. Thus there could be in principle one bipolar eruption every 30 years, and we are fully transparent in the manuscript that this higher number of eruptions should exist in reality. But the point is that nowhere nearly as many eruptions could be identified in the glacial period due to decreased measurement resolution and increased noise in the proxy archives, allowing one to only detect and match eruptions of the largest magnitudes, which are at the same time the eruptions that can be expected to have significant climate impact.

*(2c) The choice of large, 1-in-500 year bipolar events as the sole trigger for events is somewhat arbitrary. The model simulations suggest the system is most sensitive to NH forcing. So would it not make sense to use the largest NH eruptions instead?*

We agree that the eruption latitude is important for the climate response, but we are not so confident as to the exact impact of eruptions at different latitudes on the dynamics of DO cycles.
The data set of Lin et al gives a binary classification (below 40degN or Southern Hemisphere, LLSH; versus Northern Hemisphere High Latitude, NHHL) of the eruption latitude based on the relative Greenland-Antarctic deposition. We do not believe, however, that this data is informative and reliable enough to incorporate it into our statistical analysis, or even use it as the main data of the analysis.
We will however discuss it in the revised manuscript (Sec. 3.7 and Discussion), in relation to our proposed mechanism of the volcanic trigger, giving the following details:

There are 5 (LLSH) vs 2 (NHHL) eruptions within 50 years before a DO onset, and 4 (LLSH) vs 1 (NHHL) eruptions within 20 years. This is out of 34 (LLSH) vs 48 (NHHL) total bipolar eruptions.

While the sample size of eruptions before DO onsets is very small, it still indicates that eruptions classified as LLSH are more likely to cause a DO warming. We understand this could be taken as evidence that our hypothesis of a North Atlantic (NA) cooling to trigger the DO onsets is not so likely.

However, the real climatic impact and its latitudinal footprint is still very much unconstrained, since the classification of Lin et al 2022 only tells us whether there has been relatively more deposition of sulfur in Greenland versus Antarctica.

The first obvious impediment is the large uncertainty in the deposition estimates of individual eruptions, due to inter-core variability and other factors. But even if this uncertainty was zero, there is still a massive uncertainty in the climatic impact, because we do not know where most of the sulfur aerosols actually were transported (troposphere or stratosphere) and how long they were present in the atmosphere. So there is the possibility that an eruption with a large sulfur deposition in Greenland compared to Antarctica still produced less NA cooling than a tropical eruption classified as LLSH.

Further, the latitudinal estimate "LLSH" includes NH eruptions up to 40N. So based on this classification we do not have a clear separation of eruptions with and without a NA climate impact. Note that even tropical eruptions tend to have a hemispherically asymmetric cooling towards the NH, as discussed in the manuscript.

All in all, the latitude classification by Lin et al 2022 does not warrant us to focus the analysis on either latitudinal subset. Even more so because there is yet no other evidence for our NA cooling hypothesis that would warrant us to have the hypothesis inform the data analysis.

We will, however, include a detailed discussion on this in the manuscript (Sec. 3.7 and Discussion), so the reader can decide to what degree this is in conflict with our physical hypothesis.

*(3) The discussion of literature is not very balanced, and heavily favors self-citation over important prior work by others.*

We are happy to receive suggestions on important literature that was overlooked, and will add some references in the Introduction to other prior work regarding the influence of external forcing on DO cycles, as detailed further below. We still believe that all existing self-citations are appropriate as they are directly related to our study and cannot be replaced by others, as explained in the following.

Works with primary contributions by the authors:
Svensson et al 2020 and Lin et al 2022 are crucial because these are the two datasets that this study is based on.
Lohmann and Ditlevsen 2019: We use the method present in this paper to estimate the stadial onset times.
Lohmann and Ditlevsen 2021: This is crucial because it is the first that uses the Veros model (besides a benchmarking study that is also cited), and it contains methodology and further information on the tipping point of the AMOC in this model.
Lohmann 2019: This is to our knowledge the only paper that shows actual predictability of DO onsets. There exist other papers on e.g. early-warning signals of DO events, but these have no predictive power, and the observed signals may also arise due to other reasons. Since the fact that DO events are predictable to some degree at first glance contradicts the idea of a volcanic trigger, the paper needs to be discussed.
Lohmann and Ditlevsen 2018: This paper is used to highlight that there is evidence for an external modulation of DO cycles. Here we agree that other studies may be cited, which evolve around the same issue. We will add a few important studies on data analysis and data-driven modeling (Buizert/Schmittner Paleoceanography 2015; Kawamura et al. Sci. Adv. 2017; Mitsui/Crucifix Clim Dyn 2017), as well as Earth system modeling (Brown/Galbraith Clim Past 2016; Zhang et al Nature Geosc. 2021; Kuniyoshi et al GRL2022; Vettoretti et al Nature Geosc 2022).

Other self-citations with A. Svensson are mainly standard references to the underlying datasets:
Svensson et al 2006, 2008, Seierstad et al 2014: standard references for the GICC05 time scale.
Rasmussen et al 2014 (definition of Greenland stadials and interstadials, including estimates for their duration); Rasmussen et al 2013 (NEEM matching to GICC05); NGRIP members 2004 (NGRIP isotope data; Gkinis et al 2021 (NEEM high-resolution isotope data).

Further, there are these studies with A. Svensson as co-author:
Abbott et al 2021: Most recent study on the Younger Dryas volcanic hypothesis.
Capron et al 2021: One of the recent studies that investigate the timing of the onsets at decadal precision. We also cite all others that we are aware of.
Schüpbach et al 2018: Important reference regarding the impurity signals in ice cores during the glacial.

*(4) Work by Sigl et al. (2015) suggests that the largest volcanic eruptions influence climate by up to 10 years. Why do you use a 50 year threshold instead? Is there a basis for this number?*

We do not use a 50 year threshold, but investigate all lags from 10 to 70 years. The specific numbers of eruptions using a 20-year and 50-year tolerance are simply used in order to present concrete numbers to the readers, allowing them to judge the results depending on whether they a priori believe in only a short-term (20-year) or more longer-term (50-year) impact of eruptions.

Sigl et al 2015 look at the response of tree ring data, which mostly capture atmospheric temperature in regions with temperature-limited tree growth. A mechanism that triggers abrupt climate change from volcanism in the last glacial period is likely to involve other feedbacks than the immediate tropospheric cooling, which are not necessarily captured in the tree ring signal. There are numerous modeling studies (e.g. Pausata et al., PNAS 2015; Stenchikov et al., JGR 2009; Swingedouw et al., Nature Comm 2015) that show that the impact of a large eruption, especially on the ocean circulation, can last for longer than 10 years.

*P1 L16: remove "past".*

Ok.

*P2 L5-10: this seems contradictory. The models now show unforced oscillations, so wouldn't this obviate the need for triggers / drivers?*

Indeed, at first glance this may seem contradictory. But we discuss why the notions of unforced, deterministic DO cycles and a short-term volcanic trigger are not mutually exclusive at various places in the manuscript (P2 L17-22, P15 L16-22, P16 L1-3, P17 L6-9, P17 L11-17, P19L35ff).

Based on previous evidence regarding studies on the predictability of DO events (Lohmann GRL 2019) and self-sustained oscillations obtained in models (Brown/Galbraith Clim Past 2016; Vettoretti/Peltier GRL 2016, and others), we interpret our finding (the statistical relationship of eruptions and DO events) such that a trigger is not necessary for DO events to occur, but may still operate to lead to a transition that occurs earlier than otherwise (or potentially also a delayed transition in case of the interstadial-stadial transition).
However, at this stage this is only a hypothesis, and the point of our statistical analysis is to precisely assess whether there is evidence for a trigger of the events, regardless of whether a priori other evidence points towards the fact that DO events are an unforced, deterministic phenomenon.

Note that it is not certain whether the current models are in the correct dynamical regime. At different boundary conditions, the dynamics may be excitable for instance, where a driver would be needed while the same physical mechanisms are at play. This is why in this text passage we also state that there is no consensus whether there are any concrete drivers of DO ovens, or whether there is a need for any drivers. Studies that investigate the effect of volcanic eruptions on self-sustained DO-like cycles in comprehensive models are underway.

*P2 L12: Though this depends really on the subjective choice of what one decides to call a stadial or interstadial. The events that follow the traditional numbering are all over 1000 years.*

Indeed, but we would argue that the traditional numbering is based on fewer and lower-resolution records. We follow the widely applied stratigraphic framework of Rasmussen et al, QSR, 2014.

The shorter interstadials therein are now confirmed also in speleothem records (see e.g. NALPS record), so it is not subjective that these short events exist. If they should be called GI/GS or something else would depend on whether they are actually created by a different mechanism, which is not known at this point. In fact, our study contributes to make progress regarding this aspect.

*P2 L15-L22: Please refrain from self-citation in favor of a balanced review of the literature. For example, a long literature exits on the dependence of the DO timing characteristics on background climate conditions that predates the work by the author himself.*

Ok, see reply above.

*P3 L10: Is there a published bipolar ice core chronology? Or just the volcanic ties?*

It is just the bipolar volcanic match points in the cores that are available. These are used to transfer the Greenland GICC05 chronology to the Antarctic ice cores. Will rephrase the sentence.

*P4 L3: Are the*

Not sure what is meant here.

*P6 L10-15: "Thus, most eruptions in the dataset of Svensson et al. (2020) fall into the category of 1-in-500 year events in terms of their magnitude". This is unclear to me. Do the 82 bipolar events you identify overlap with the 69 events by Lin 2021?*

*Further down it seems this is addressed in Section 3.2. Please consolidate this information into one place to avoid confusion. Although section 3.2 refers back to 2.3, so it is not clear how the analysis was done.*

Yes the 69 events by Lin 2022 are a subset of the N=82 bipolar set in Svensson 2020. We will add a Section 2.2. where we more clearly describe the two volcanic data sets and how they relate.

*P6 L16: I strongly disagree with this conclusion. The Svensson (2020) paper itself clearly states that it is incomplete: "The bipolar layer counting is not continuous but is focused on periods of abrupt climate variability or high volcanic activity."*

See response above.

*P6L 22: what do these p values signify? What threshold do you use?*

I am not sure what is meant here with thresholds, but for instance a p-value below 0.05 would signify that the empirical distribution of the waiting times is inconsistent with an exponential distribution at 95% confidence. The statistical tests used are standard. Since we clearly find p>0.1 (or whatever confidence level one prefers), there is no evidence that the data is inconsistent with a Poisson process.

*P6 L32: It seems clear that the deglaciation ( 12-17ka) and glacial (24-60 ka) have different volcanic statistics. It is true one expects 2 false positives at 90% confidence, but that ignores the fact that BOTH of these are consecutive, and BOTH are in the youngest segment with thicker annual layers and thus higher detection probability for volcanic layers.*
*A t-test of waiting time distributions on either side of the data gap would probably suggest these are different.*

We agree that it is likely that this is a systematic effect and also state this on page 6. But from the point of statistics, we do not have enough evidence that this is really the case. It is thus not reasonable to construct our statistical framework in a way that would somehow take this into account.

For more, see our more detailed response to your comment further above.

The fact that the intervals in question are (almost!) consecutive will not really change the statistical (non-)significance, since this indicates that there is a (plausible) correlation in the occurrence frequency of 2 kyr segments, which would reduce the number of effective data points in the multiple testing context.

*Figure 3: Do you use the estimates from Lin 2021 for the bipolar volcanoes? In Fig 3c, what is the aerosol loading based on?*

Yes, Lin 2021 is the only available source for the deposition estimates. This is stated in the figure caption. See Lin 2021 for more details on the calculation of the aerosol loading.

*P11 L8: I don't understand the logic. Antarctica also has several large local volcanoes in close proximity (on the continent). Also, these NH eruptions probably had more impact on the AMOC that you care about.*

The logic is the following:
Since the Greenland data set contains more eruptions than that of Antarctica, and a bipolar match requires an eruption in both Greenland and Antarctica, there are by definition more "local/regional/unipolar" eruptions in the Greenland data set. One could spin this the other way around and say that simply more eruptions are missing in the Antarctic records.
But neither interpretation would influence our analysis since then also the dataset in Svensson 2020 (based on the same ice cores) would be missing these eruptions, as they require deposition in both Greenland and Antarctica, and this would be the case regardless of whether they occurred close to DO onsets or not. Thus, as already stated in the manuscript, it is logical to use the smaller Antarctic dataset to make our inference on the potential number of missing bipolar candidates.
We agree that there are also local eruptions in Antarctica. There is no reliable data yet as to whether Greenland or Antarctica features more "local" eruptions, whatever the definition thereof. We will rephrase to not give the impression that there are only local eruptions close to Greenland.

Regarding your last point, we don't believe we have reliable data yet to choose a subset of eruptions for our analysis that we think has a priori more impact on the AMOC. See also our responses to your previous points regarding the latitudinal issue.

*P11 L10: Out of these 84 largest eruptions, how many correspond to bipolar events?*

None. The point of this analysis is to find all eruptions with a large sulfur deposition that have not been identified as bipolar by Svensson 2020. This will include some that actually had a bipolar imprint, but also plenty of eruptions that only have a large sulfur deposition because the eruption site was close to the ice coring site. That is why we say this is an upper bound estimate.
Since this has not been brought out clearly enough in the manuscript, we will add a Section 2.2., where the relationship of the Lin et al 2022 and Svensson 2020, as well as the unipolar subset of Lin et al 2022, is addressed explicitly.

*P13 L20: Personally I think at 50 years it would already be unlikely. Sigl et al. 2015 shows a ~ 10 yr impact on climate of the largest eruptions.*

As pointed out above, Sigl et al 2015 only show the impact on atmospheric temperature in tree ring records. The following response of the ocean circulation could certainly take decades before an actual regime shift of the circulation starts, and it is not so clear how this would be manifested in tree ring data. Leads and lags in between the eruptions and the DO onsets below 15 years are unfortunately hard to assess as this is within the uncertainty of the onset detection.

*Section 3.5: Figure 5b is hard to interpret. Fig. 5a is much more intuitive. Can you also plot the result when doubling the events there?*

We agree that panel a is easier to understand, but it also contains less information, since from panel a one cannot infer the significance of the results in a quantitative way. Because of the discrete nature of the distributions, the confidence bands are only defined to the nearest integer (being conservative, here we choose the next larger integer).
We will improve our explanation of panel b so that it is immediately clear what is shown. It is simply the p-value of our null hypothesis as a function of the free parameter.

Thus, we will include the expectation value of the doubled occurrence rate in panel a, but not the confidence bands. The latter would be misleading as they do not allow one to judge whether the data is really significant at 90% or 95% confidence in case the data is close to these bounds, as is the case here for the doubled occurrence rate (see Fig. 5b).

*Section 3.7: It is not surprising that your model is more sensitive to NH cooling. Does this not imply that you should be evaluating the proximity of DO event onsets to large NH volcanoes?*

It may not be surprising in relation to our physical hypothesis, but it still not clear a priori that this is indeed the case in a model. As stated previously above, we believe our statistical analysis is more powerful when not constrained by a preconceived hypothesis on a particular subset of eruptions. Otherwise we might be accused of pursuing a preformed hypothesis.

As also stated above, this is especially true since we do not actually have reliable information on the latitude of the eruptions, and even less on their true climatic impact on the global or NH climate.

*P19 L5: My main concern is the incompleteness, and the fact that the focus of the identification of bipolar events focused on periods of abrupt climate change.*

This is exactly what we mean here. In the revised manuscript we will address the issue of the potential systematic bias around DO events explicitly, including providing more evidence to the contrary, as discussed above.

*P20 L11: These kind of statements are somewhat tentative, as this depends a lot on the weather conditions that distributed the volcanic deposits to both polar regions, and depositional processes and redistribution on the ice sheet surface.*

Unfortunately it is not clear which statement this refers to, and what the referee want us to change.

*P20 L12: "arguably more accurate"? I would say certainly more accurate!*

Yes in a way they are clearly more accurate, but in another way they are not: Observations of 2,500 years do not give a precise estimate on the magnitude of a 1-in-500 year eruption.

*P20 L19: I believe this is in violation of the data policy; all data should be made available. I will let the editor weigh in.*

We will leave this up to the editor.

**Referee #4**

Below are our responses to the issues raised by the referee (*italic*).

*Potential biases in the timing of identified bipolar matches. The bipolar matches in Svensson et al. (2020) represent a considerable effort, but it's worth remembering how uncertain the matches are; there are few definitive linkages between volcanic events - such as tephra or even sulfate isotopes that indicate stratospheric eruptions. Which is not to say they should not be used in the manner in this manuscript, but considerable caution should be applied. Because the matched volcanic events rely upon pattern matching sulfate peaks with a similar number of annual layers between, the identification of matches may be biased to when the timescales are already well synchronized – the abrupt DO events. It would be good to include a discussion of how many volcanic events fall in the 50 years after a DO-warming.*

We thank the referee for this suggestion and will include this in the manuscript (Sec. 3.1 and new panel in Fig.2). In essence, there is no DO warming where an eruption occurs within 25 years after the onset, and 2 DO warmings where this is the case within 50 years. Under the null hypothesis we would expect this to happen by chance for 1 event within 25 years and 2 events within 50 years. This is opposed to 5 events with eruptions within 20 years and 7 events with eruptions within 50 years when looking before the onset. Because of this and other reasons already stated in the manuscript, we are fairly confident that the increased frequency of eruptions before the onsets is not an artifact of the better prior matching of the records close to the DO transitions.

*Another issue for this work is whether the identified volcanic matches are accurate. As the recent GICC revision (GICC21 for the past 3.5ka, Sinnl et al., 2022) shows, mis-identification is not a trivial problem. The paper addresses well whether bipolar eruptions are underestimated overall; but it does not address the number of misidentifications. This seems like the largest uncertainty to me. There is a volcanic eruption identified every 50 years, most of which do not reach both poles. So there are likely to be a significant number of instances where there was an eruption in both the NH and SH within a few years of each other that could be mis-identified as a bipolar eruption (accuracy of annual layer interpretation in the glacial for both GICC05 and WD2014 is probably close to 10% on shorter intervals, which greatly increases the number of events that could be considered coincident).*

It is correct that the problem of misidentified bipolar eruptions is not addressed in the manuscript. This is because a) it is very hard to quantify this in a rigorous way, and b) we nevertheless believe the number of potentially faulty matches is relatively low, as detailed below.

The probability of a bipolar match occurring just by chance is hard to estimate, because the bipolar matching is a quite complicated procedure, relying on information from many different records, and it was not done in the exact same way for all eruptions. Still, our analysis of the data in Lin 2022 may be used to give a rough estimate of how likely it would be to find bipolar matches by chance. Since we have estimates for the magnitudes from Lin 2022, we can see that while as you say there are eruptions identified every 50 years at the individual poles, eruptions with a deposition magnitude corresponding to the bipolar ones in Svensson 2020 only occur roughly every 250 years, as discussed in the paper. Now for a bipolar match one would actually need at least two consecutive eruptions that coincide in Greenland and Antarctica within a given age tolerance, and which are above this magnitude threshold. This greatly reduces the likelihood of coinciding eruptions, as follows:

Assuming the simplest of such patterns (2 consecutive eruptions), the first step is to find an eruption in Greenland and Antarctica that lies within the uncertainty of the prior methane or 10Be matching, which is around 100 years around the time of the DO events (see Fig. S16 in Svensson 2020). From the Poisson process model, this gives a probability of a misidentification (i.e. an eruption coinciding in Greenland and Antarctica within the uncertainty window just by chance) of $P\_1 = 1 - \exp(-100/250) = 0.33$.
In the second step, another eruption is identified at both poles, and it is checked by layer counting whether the number of years elapsed from the first eruption is almost equal in both poles. Here "almost equal" means "within the relative layer counting uncertainty", which as you say could be estimated by 10% of the counted interval.
Assuming this second eruption is spaced in one of the poles by exactly the expectation value 250 years (of course more rigorously this spacing is a random variable with exponential distribution), the probability of finding the second eruption in the other pole within the window admissible by the counting uncertainty is: $P\_2 = 1 - \exp(-0.1*250/250) = 1 - \exp(-0.1) = 0.095$.

These probabilities have to be multiplied to give an estimate of the probability of a misidentified doublet: $P = P\_1 * P\_2 = 0.03$, or 3%. For a total of N=82 eruptions (should be strictly less, since the eruptions are identified in patterns and not individually), this would give an expected value of 2.58 misidentified eruptions.

Note that even if one would for some reason doubt our estimated recurrence time of candidate eruptions of 250 years, and use a smaller time instead, the overall probability does not become higher than $P = 1- \exp(-0.1) = 0.095$ or 9.5%, yielding a maximum expected value of 7.8 misidentifications. This is because, while P\_1 will approach 1, P\_2 remains unchanged (since the counting uncertainty is proportional to the time in between eruptions). Note that for patterns longer than 2 eruptions (as was the norm in Svensson 2020) these probabilities will become much smaller.

We don't claim this is in any way a precise estimate of the actual probability of false bipolar matches in our data set. But we just wanted to give some arguments as to why we believe misidentified patterns of eruptions are much less common than one might think at first, and are most likely rare enough so that they will not influence our analysis and results.

*2) Magnitude of identified events at the DO warmings. I looked up the 7 volcanic events at the DO warmings in Table 2 of Lin et al. 2022. I was surprised that only 3 of these were in the top 45 largest magnitude, and only one was a Northern Hemisphere eruption. But maybe most surprising that two of warming with volcanic events had much larger volcanic events that preceded them by decades to a century (14761 was the 21$^{st}$ largest and 38366 was the 39$^{th}$ largest). In both cases, the stadials were already long and stable, which raises the question of why the larger events did not trigger a DO warming? I also looked quickly at the largest event (55383) which occurs during a time of "flickering", suggesting the climate was susceptible to external forcing. Yet it did not produce a DO warming (possibly a cooling?). The manuscript would benefit from providing more context on the magnitude of the identified volcanic forcing and how it compares to other volcanic forcing that preceded, but did not trigger, a DO warming.*

Indeed, when looking into single events, like the ones suggested by the referee, one may wonder why a DO event was not triggered earlier by another event. One way to interpret why there are some larger eruptions that do not trigger a DO onset, while smaller eruptions shortly after do, is that the larger eruption already partially destabilized the system, and thus a smaller eruption was sufficient to trigger the transition. Of course, this is completely speculative, but it serves to illustrate that there is no fundamental dilemma here.

Apart from this, we find it difficult to assess the significance of eruptions that did not trigger DO events in a meaningful, quantitative way. There are obviously many more "non-triggering" eruptions, so one would have to find an objective criterion of an eruption that "should" have been a trigger. We don't see an obvious suitable statistical framework to devise here, but we are happy to receive suggestions.

Since the philosophy of the work is a robust statistical analysis, we would like to refrain from pointing towards individual eruptions in order to make any claims. Not least because it is highly uncertain whether individual eruptions, such as the ones you mention, are really larger compared to others, due to the large uncertainties in the deposition estimates from Lin 2022 and even more so in the actual climatic impact on the relevant parts of the climate system, which is fully unconstrained.

Regarding the magnitude ranks you mentioned, we show in the manuscript that the estimated deposition of the eruptions before DO events is slightly smaller on average compared to the whole bipolar population, but not significantly so. It may be that you are surprised to only find 3 out of 7 eruptions in the top 45, but assuming that the 7 eruptions are an unbiased random sample from the N=82 eruptions the two most likely outcomes are that you will find 3 or 4 eruptions larger than the 45th largest eruption, since eruption number 45 is very close to the median of the sample. Further, an eventual bias towards too small eruptions is already taken into account when testing for the robustness of our results to undercounting.

Regarding your observation of only one eruption being classified as "Northern Hemisphere":

The data set of Lin et al indeed gives a binary classification (below 40degN or Southern Hemisphere, LLSH; versus Northern Hemisphere High Latitude, NHHL) of the eruption latitude based on the relative Greenland-Antarctic deposition. We do not believe, however, that this data is informative and reliable enough to incorporate it into our statistical analysis or to warrant a specific re-interpretation of our results. We will however discuss it in the revised manuscript, also in relation to our proposed mechanism of the volcanic trigger, giving the following details:

There are 5 (LLSH) vs 2 (NHHL) eruptions within 50 years before a DO onset, and 4 (LLSH) vs 1 (NHHL) eruptions within 20 years. This is out of 34 (LLSH) vs 48 (NHHL) total bipolar eruptions. While the sample size of eruptions before DO onsets is very small, it still indicates that eruptions classified as LLSH are more likely to cause a DO warming. This could be taken as evidence that our hypothesis of a North Atlantic (NA) cooling to trigger the DO onsets is not so likely. However, the real climatic impact and its latitudinal footprint is still very much unconstrained, since the classification of Lin et al 2022 only tells us whether there has been relatively more deposition of sulfur in Greenland versus Antarctica. But there is a large uncertainty in the deposition estimates of individual eruptions, due to inter-core variability and other factors. Even if this uncertainty was zero, there is still a massive uncertainty in the climatic impact, because we do not know where most of the sulfur aerosols actually were transported (troposphere or stratosphere) and how long they were present in the atmosphere. So there is the possibility that an eruption with a large sulfur deposition in Greenland compared to Antarctica still produced less NA cooling than a tropical eruption classified as LLSH.

Finally, note that the latitudinal estimate "LLSH" includes NH eruptions up to 40N. So based on this classification we do not have a clear separation of eruptions with and without a NA climate impact. Further, even tropical eruptions tend to have a hemispherically asymmetric cooling towards the NH, as discussed in the manuscript.

*Figure 2. I would like this figure to include the volcanic events that occur after the DO warming. This could illustrate the volcanic events are more like to precede the climate transition than to follow it, which would support the inference of causality.*

We agree and will add a panel b) where the same is shown for eruptions occurring after the DO warming onsets. This will make it visually very clear that there is an obvious asymmetry in between eruptions occurring before and after the onsets, which indeed suggests a causation from eruptions to DO warmings.

*It seems like some of the information taken from Lin et al. needs updating, likely due to changes in the review process for that manuscript.*

Indeed some minor details have changed since our submission, mostly in the Antarctic deposition estimates. The analysis is now updated according to the published dataset, and all figures updated. The results are unchanged, and specifically the very small changes in the deposition estimates had no discernible influence on the upper bound of the eruption occurrence rate.

*The climate model used seems too simplistic. I get that it is a toy model to show plausibility, but more justification for why the model has the important components to address this issue would be helpful. There are only 3 references in section 2.4. Maybe the benchmarking of the model occurs in Lohmann and Ditlevsen, 2021, but if so, some of the relevant content should be repeated.*

First, we would like to mention that the model used is a fully fledged ocean-only GCM and not a toy model. In the paper we do acknowledge that it is coarse resolution and uses present-day boundary conditions. We do not claim it is a realistic model that can explain DO events, and the purpose of our manuscript is not to provide a full explanation of how volcanic eruptions would influence DO events. The main purpose is to show that a statistical association exists in between volcanic eruptions and some rapid DO warmings, by means of an analysis of the ice core data that does not depend on any physical hypothesis or modeling. We then provide an admittedly speculative explanation of why the eruptions may preferably trigger DO warmings and not coolings, which is made plausible by model simulations.

The only really important component that the model needs to fulfill (besides being a three-dimensional ocean model with global bathymetry) to show the plausibility of our hypothesis is to have a tipping point of the AMOC, along a regime of bi-stability. For the latter, one needs to have an ensemble of equilibrium simulations, going beyond the usual hysteresis experiments.
There are actually not many models of higher complexity that would fulfill these criteria. Since, as we state in the manuscript, the model does not have an active sea ice or atmospheric component, we cannot assess how sea ice or atmospheric dynamics may alter the plausibility of our mechanism. The mechanism is thus contingent on the sea ice and atmospheric response to not destroy the bi-stability of the AMOC and the dense water formation in the North Atlantic as a volcanic trigger of an AMOC transition. This will need to be tested in fully coupled models that actually have a reasonable representation of DO-type dynamics, and such studies are actually underway. We will make these points more visible in Sec.'s 2.4 and 3.7 of the revised manuscript, and repeat some more crucial information on the model from Lohmann and Ditlevsen 2021.

*And a final note on authorship. Given the reliance on the data set of Lin et al., which was only recently accepted for publication (the Ides of March if I remember right), it seems like adding authors would be warranted.*

We agree that the data from Lin et al is important for our robustness tests, but since it is published we prefer to only include authors that directly contributed to this work. We are collaborating with key authors from Lin et al on follow-up studies.

---

## Author Response (AR2)

**Response to Reviewer #1**

We thank the referee for reviewing the manuscript again and for giving further instructive comments.

As suggested, we kept the discussion of Sec. 3.6 and slightly extended it to make it clear that with the current data set the question of a volcanic trigger of DO cooling events is still inconclusive. Here we specifically mention all 3 remaining issues identified by the reviewer as to why the analysis on the cooling events is limited at present, including new references to papers suggested by the reviewer that highlight the large impact that extra-tropical eruptions (not considered in our paper) can have on the climate.

A side note regarding these new references that consider the climate impact of NH extra-tropical eruptions:
In accordance to the mechanism we propose with our ocean model simulations, a large NH cooling (as suggested by these references) does not lead to a weakening of the AMOC (shutdown of AMOC = transition to a stadial) but instead a strengthening (e.g. van Dijk et al 2022). Thus, it is not clear whether such eruptions are good candidates for a stadial onset trigger. But of course we cannot say at this point that our hypothesized mechanism is the correct one.

We further removed the reference to DO coolings from the abstract, and give a more careful interpretation of our analysis in the Discussion/Conclusion section, where a mention of the cooling events has been removed in one of the two relevant passages, and the other passage has been appropriately caveated, as suggested.

**Response to Reviewer #2**

We thank the reviewer for reviewing our revised manuscript and for the useful suggestion to more clearly highlight limitations of our approach.

1.) Following the suggestion by the referee to make the main limitations of the study more prominent, we added the following to the Discussion/Conclusion Section (slightly modified from the Referee suggestion):

*"As a result, while the SVE20 bipolar volcanic catalogue certainly undercounts the true number of bipolar volcanic events of arbitrary strength, we argue that it captures a sufficiently large portion of the strongest events most relevant to triggering climate change. While our analysis only considers bipolar volcanic eruptions that have been identified in the glacial sections of the ice cores used, volcanic events restricted to either the northern or southern hemisphere may likewise contribute to abrupt climate change. However, uncertainty in assessing their latitude and magnitude precludes us from evaluating them here."*

And the following to the abstract:

*"While we argue that the bipolar catalogue used here covers a sufficiently large portion of the eruptions with the strongest global climate impact, volcanic events restricted to either the northern or southern hemisphere may likewise contribute to abrupt climate change."*

2.) Regarding the minor comment that we are severely undercounting bipolar eruptions:

We do not claim that there can be only up to 2 times more bipolar eruptions (of any size) in the record, or that 1-in-500 years is our estimate of the return period of all bipolar eruptions that would have any discernible signature in the ice cores at both poles. 500 years is simply the return period we find in the SVE20 data, which we argue is a data set with eruptions above a certain threshold in magnitude. We then find that the characteristic magnitude of these eruptions is indeed consistent with the largest eruptions of the last 2,500 years (roughly Tambora-sized) of the same return period (and not of all bipolar eruptions of any size), meaning the catalogue is likely relatively complete. Based on the unipolar deposition magnitude (LIN22 data), we then further find that there could only be up to 2 times the number of bipolar eruptions of this characteristic magnitude (sulfate deposition) during the investigated time period, assuming there would be no local eruptions with large but only unipolar sulfate deposition.

We already stated in the relevant parts of the manuscript that we are trying to constrain the number of missing bipolar eruptions of the same characteristic (large) magnitude, and that only eruptions of this characteristic magnitude are relevant to our statistical analysis. In the revised manuscript, we made this more explicit in several places. To make it more clear, we added the following to Sec. 3.2:

*"Further, the numbers given here, i.e., the return period of 500 years as well as the estimate of eruptions potentially missing from the SVE20 data set, do not refer to bipolar eruptions of any size, but to bipolar eruptions of the characteristic (large) size of the bipolar eruptions in the SVE20 data. The former are indeed known to occur much more frequently (Sigl et al 2022)."*

The previous version of the manuscript did acknowledge that there are 80 eruptions in the last 2500 years based on Sigl et al. 2015. Actually, we do not claim anywhere in the manuscript that this time period "is too short for strong conclusions". I assume this was a reference to some of our previous author responses in relation to the Rougier et al. data.
Certainly, a data set of the last 2,500 years is not suitable to give statistically robust results for eruptions with return periods on the same order of magnitude, nor is it applicable directly to the study of DO events. However, the recurrence time for bipolar eruptions (of any size) is indeed well-constrained, yielding roughly 31 years, consistent with the newer data covering the entire Holocene.
We agree of course that it is good to include the newer data, and now include the new estimate of 35 years from the extended record by Sigl et al 2022.

We further added the following clarification in Sec. 2.4:

*"By saying the data set is relatively complete, we mean that it covers a sufficiently large portion of the bipolar eruptions above a certain threshold in magnitude, which corresponds to eruptions with return periods of 1 in 500 years and larger. In contrast, we do not mean that it represents a complete catalogue of all bipolar eruptions of any size that could be detectable in more highly resolved and better synchronized ice core records, such as during the Holocene (Sigl et al 2022)."*

3.) As suggested, we added 2 panels with time series segments of the two instances to Fig. 1.